EMBO
Molecular Medicine

# DUSP6 inhibition overcomes neuregulin/HER3-driven therapy tolerance in HER2+ breast cancer

Majid Momeny[1,2✉], Mari Tienhaara [3,4], Mukund Sharma [1,4], Deepankar Chakroborty[3,4], Roosa Varjus[1], Iina Takala [3,4], Joni Merisaari[1], Artur Padzik[1], Andreas Vogt [5], Ilkka Paatero [1], Klaus Elenius [3,4], Teemu D Laajala [6], Kari J Kurppa [3,4] & Jukka Westermarck [1,4✉]

## Abstract

**Despite clinical benefits of tyrosine kinase inhibitors (TKIs) in cancer, most tumors can reactivate proliferation under TKI therapy. Here we present transcriptional profiling of HER2+ breast cancer cells transitioning from dormant drug tolerant cells to re-proliferating cells under continuous HER2 inhibitor (HER2i) therapy. Focusing on phosphatases, expression of dual-specificity phosphatase DUSP6 was found inhibited in dormant cells, but strongly induced upon regrowth. DUSP6 expression also selectively associated with poor patient survival in HER2+ breast cancers. DUSP6 overexpression conferred apoptosis resistance, whereas its pharmacological blockade prevented therapy tolerance development under HER2i therapy. DUSP6 targeting also synergized with clinically used HER2i combination therapies. Mechanistically DUSP6 is a positive regulator of HER3 expression, and its impact on HER2i tolerance was mediated by neuregulin-HER3 axis. In vivo, genetic targeting of *DUSP6* reduced tumor growth in brain metastasis model, whereas its pharmacological targeting induced synthetic lethal therapeutic effect in combination with HER2i. Collectively this work demonstrates that *DUSP6* drives escape from HER2i-induced dormancy, and that DUSP6 is a druggable target to overcome HER3-driven TKI resistance.**

**Keywords** Non-genetic Drug Tolerance; Lapatinib; Neratinib; BCI; FOXM1
**Subject Category** Cancer

## Introduction

To develop therapeutic resistance, tumor cells undergo distinct evolutionary stages, starting with induction of dormancy and non-genetic drug tolerance, followed by epigenetic changes and finally resistance-conferring genetic mutations (De Conti et al, 2021; Hata et al, 2016; Marine et al, 2020). These different phases were originally demonstrated for EGFR-targeted therapies in non-small cell lung cancer (NSCLC) cells (Sharma et al, 2010), but the concept has been expanded more recently to other malignancies including HER2+ breast cancer (Chang et al, 2022; De Conti et al, 2021; Dhimolea et al, 2021; Hata et al, 2016; Kurppa et al, 2020; Sharma et al, 2010). Based on these studies, there is ample of omics data from dormant cells (also called as drug tolerant persisters; DTPs) from different cancer types treated with variety of therapies. However, our understanding of the molecular mechanisms behind the regrowth of DTPs under continuous therapy is still rudimentary. Especially, to our knowledge there are no published studies describing transcriptional landscapes of transition from DTP to drug tolerant expanding cells (DTEP), or transition of DTEP cells to long-term resistant (LR) cells upon TKI therapies.

The human epidermal growth factor receptor 2 (HER2; encoded by *ERBB2*) is overexpressed in ~15–25% of human breast cancers and associates with a poor patient survival (Arteaga and Engelman, 2014; Haikala and Janne, 2021). HER2 belongs to the ERBB family of receptor tyrosine kinase (RTK) with four members: HER1 (EGFR), HER2, HER3, and HER4. Upon ligand binding, the ERBB receptors homo-and heterodimerize and activate downstream signaling pathways including PI3K/AKT and RAS/MAPK/ERK, which regulate cell proliferation, survival, and the metastatic dissemination (Arteaga and Engelman, 2014; Haikala and Janne, 2021). Multiple HER2-targeted therapies, including the monoclonal antibody trastuzumab and small molecule tyrosine kinase inhibitors (TKIs) have been approved for the treatment of HER2-overexpressing (HER2+) breast cancer (Goutsouliak et al, 2020). Application of anti-HER2 agents in combination with chemotherapy has significantly improved the patients' outcome. However, patients initially responsive to the HER2 inhibitors (HER2is) almost inevitably succumb to disease relapse (Goutsouliak et al, 2020). Moreover, HER2+ breast tumors have an inherent tendency to develop brain metastasis, a significant clinical challenge for the treatment of these patients (Fecci et al, 2019). Therefore, there is a pressing need for novel and more efficacious therapeutic strategies to overcome resistance to HER2is.

HER3 is an obligate heterodimerization partner for HER2 and plays essential roles in HER2-driven tumorigenesis, and resistance

[1]Turku Bioscience Centre, University of Turku and Åbo Akademi University, Turku, Finland. [2]The Brown Foundation Institute of Molecular Medicine, McGovern Medical School, The University of Texas Health Science Center at Houston, Houston, Texas, USA. [3]Medicity Research Laboratories, Faculty of Medicine, University of Turku, Turku, Finland. [4]Institute of Biomedicine, University of Turku, Turku, Finland. [5]University of Pittsburgh Drug Discovery Institute, Department of Computational and Systems Biology, Pittsburgh Technology Center, Pittsburgh, PA, USA. [6]Department of Mathematics and Statistics, University of Turku, Turku, Finland. ✉E-mail: majid.momeny@uth.tmc.edu; jukwes@utu.fi

to HER2is (Haikala and Janne, 2021; Wilson et al, 2012). Consistent with the effects of HER3 overexpression, the HER3 ligand neuregulin (NRG, a.k.a Heregulin; HRG) promotes trastuzumab resistance in HER2+ breast cancer cells (Haikala and Janne, 2021). Importantly, the NRG-HER3 axis also promotes resistance to a wide range of TKIs and chemotherapies (Erjala et al, 2006; Haikala and Janne, 2021; Knuefermann et al, 2003; Recondo et al, 2020; Wilson et al, 2012; Yonesaka et al, 2011). Despite the importance of HER3 in cancer progression and therapy resistance, development of HER3 small molecule inhibitors has been challenging due to its impaired kinase activity (Haikala and Janne, 2021; Xie et al, 2014). Moreover, the clinical activity of HER3 monoclonal antibodies either as monotherapies or in combination with chemo- and targeted therapies have been marginal (Cleary et al, 2017; Haikala and Janne, 2021; Schneeweiss et al, 2018). To this end, there is a pressing need to identify novel strategies to inhibit HER3 activity and/or expression for the treatment of HER3-dependent human malignancies (Gaborit et al, 2015; Haikala and Janne, 2021; Xie et al, 2014).

There is emerging evidence that phosphatases are novel and "druggable" targets in oncology (Lazo et al, 2018; Vainonen et al, 2021). Inhibition of oncogenic phosphatases, or reactivation of tumor suppressor phosphatases, by small molecule therapies halt tumor growth, retard malignant progression, and enhance therapeutic sensitivity in various neoplasms (Lazo et al, 2018; Vainonen et al, 2021). Despite this, the contribution of phosphatases to resistance to HER2is is still poorly understood. Dual-specificity phosphatases (DUSPs) belong to the superfamily of protein tyrosine phosphatases and dephoshorylate both tyrosines and serines or threonines. A subgroup of the DUSPs are mitogen-activated protein kinase (MAPK) phosphatases that selectively interact with and dephosphorylate the MAPKs (Patterson et al, 2009; Zandi et al, 2022). For instance, DUSP6 displays a high degree of substrate selectivity for the extracellular signal-regulated kinase (ERK), but not P38 or c-Jun N-terminal kinase (JNK) (Zandi et al, 2022). However, recent studies indicate that at least some cancer relevant DUSP6 functions may by ERK-independent (Kong et al, 2023). DUSP6 is indicated in clinical cancer progression, and its genetic inhibition prevents tumor cell growth (Shojaee et al, 2015; Wu et al, 2018). Notably, DUSP6 is a druggable phosphatase (Korotchenko et al, 2014; Molina et al, 2009; Vainonen et al, 2021; Zandi et al, 2022). The best characterized DUSP6 inhibitor molecule BCI ((E)-2-Benzylidene-3-(cyclohexylamino)-2,3-dihydro-1H-inden-1-one), is a semi-allosteric inhibitor of both DUSP1 and 6 that phenocopies genetic DUSP6 inhibition in several cancer models (Kong et al, 2023; Korotchenko et al, 2014; Molina et al, 2009; Vainonen et al, 2021). However, the role and potential of therapeutic targeting of DUSP6 in overcoming resistance to HER2is is currently unknown.

Here we present transcriptional analysis of HER2i-treated HER2+ cancer cells upon 9 months of continuous HER2i treatment. In addition to revealing first global gene expression programs associated with the DTP-DTEP and DTEP-LR therapy tolerance transitions, we use complementary genetic and pharmacological approaches to demonstrate that *DUSP6* has critical role in regrowth of DTEP cells under HER2i therapies. Mechanistically, DUSP6 drives non-genetic HER2i tolerance via regulation of HER3 expression, and by abrogating neuregulin-elicited apoptosis resistance. Collectively our findings provide a strong pre-clinical

rationale to further advance in DUSP6 blockade for HER3 targeting in general, and especially for the clinical management of HER2+ breast cancer patients with resistance to HER2i.

# Results

## Development of acquired HER2i resistance by long-term treatment of drug sensitive HER2+ breast cancer cells with lapatinib

To model the full range of development of HER2i resistance starting from primary sensitive phase, via drug-tolerant persister (DTP) development, and re-emergence of proliferative drug-tolerant expanded persister (DTEP) cells, to long-term resistant cells (LR), the HER2i sensitive cell line BT474, and its brain seeking variant BT474Br (Zhang et al, 2013), were exposed to therapeutically relevant 1 μM of lapatinib every 3 days for up to 9 months (Fig. 1A). Both cell lines followed a similar pattern of lapatinib tolerance development, where at the 9-day timepoint only a few DTP cells could be microscopically observed, whereas the emergence DTEP population took about 6 months (Fig. 1A). Following this, the plates were fully populated by the LR cells after 9 months of continuous lapatinib treatment (Fig. 1A). Importantly, in addition to lapatinib, the LR clones of both BT474 and BT474Br displayed strong cross resistance to tucatinib (a HER2i), afatinib (a HER2/EGFR inhibitor), and neratinib (a HER2/HER4/EGFR inhibitor) (Fig. EV1). This indicates that the acquired resistance is not specific to lapatinib but is driven by a mechanism that is generally relevant to the ERBB family of RTKs.

## Transcriptomic landscape of acquired lapatinib resistance in BT474 cells

Recent studies have focused on the molecular characterization of DTP cells in response to kinase inhibitor therapies (Chang et al, 2022; Kurppa et al, 2020; Marsolier et al, 2022) but there is no published information about transcriptional profiles of TKI-treated cells at the DTP-DTEP or DTEP-LR transitions. To this end, the transcriptional profiles from each functional state of lapatinib drug tolerance and resistance development in BT474 cells were surveyed by bulk RNA-sequencing. From three technical replicates per condition, and by using statistical criteria of $|logFC| > 2$ and Benjamini-Hochberg adjusted $p < 0.05$, upregulation of 144, 1169 and 16 genes was found upon the control-DTP, DTP-DTEP, and DTEP-LR transitions of BT474 cells, respectively (Fig. 1B, Dataset EV1). On the other hand, the number of downregulated genes were 517, 930, and 28, respectively (Fig. 1B, Dataset EV1). The highest number of differentially regulated genes upon the DTP-DTEP transition indicates that this is the transition phase where the BT474 cell fate is most robustly impacted during the resistance development. When assessing the patterns of gene expression changes by unsupervised soft clustering analysis (Futschik and Carlisle, 2005), we identified six approximately similar size gene clusters with distinct regulation patterns (Appendix Fig. S1A). The genes included in these clusters are listed in the Dataset EV2. Such regulation patterns indicate that neither gene activation nor gene repression characterize lapatinib resistance development in BT474 cells, but unique gene expression programs are involved in each of these steps.

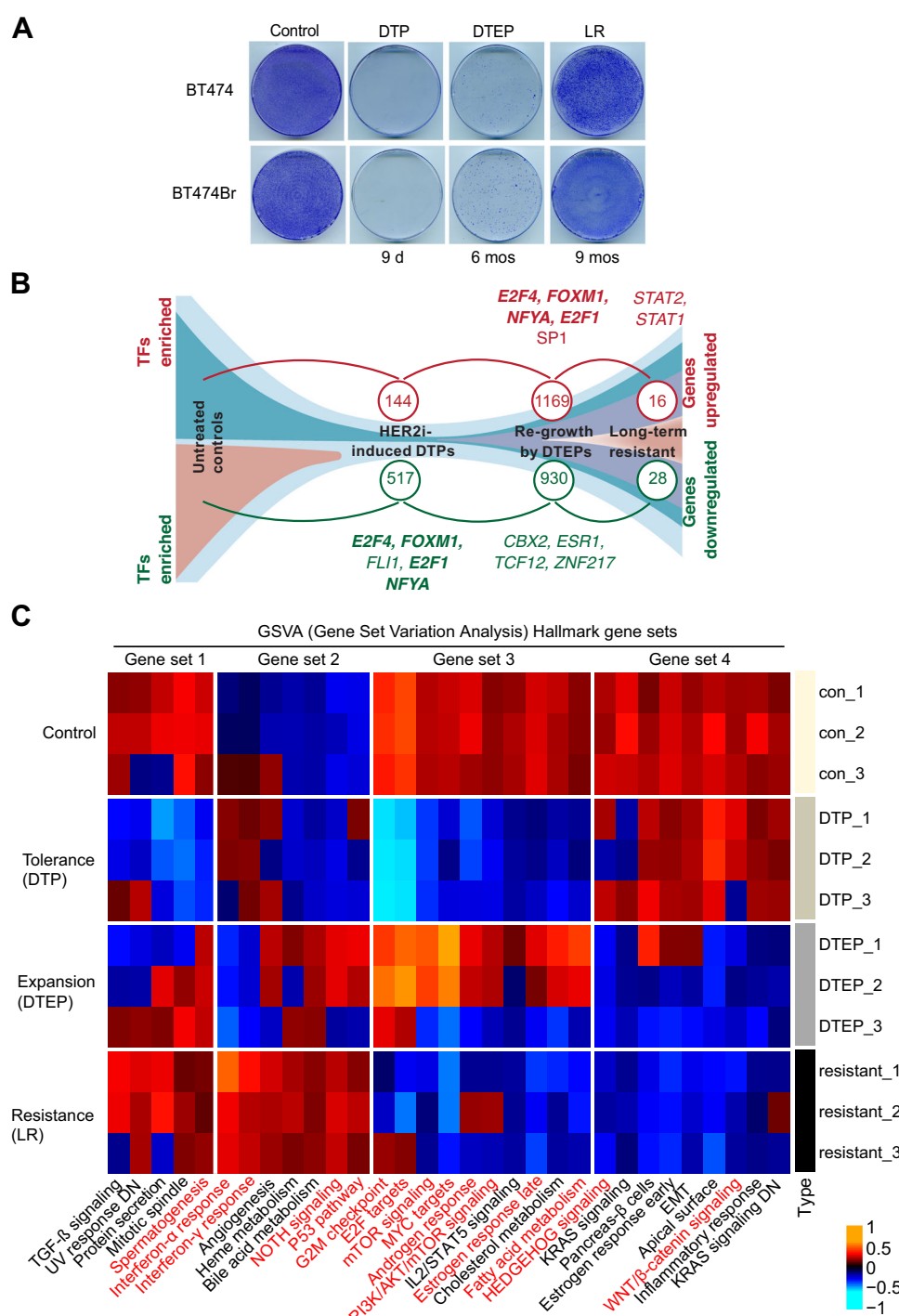

**Figure 1.  Transcriptional landscape of lapatinib tolerance and resistance development in HER2+ cells.**

(A) Development of lapatinib resistance in HER2+ breast cancer cells. BT474 and BT474Br cells were treated with 1 µM of lapatinib for 9 days (d), 6 months (mos) and 9 mos to yield drug-tolerant persister (DTP), drug-tolerant expanding perister (DTEP), and long-term resistant (LR) clones, respectively. The cells were stained with crystal violet (0.5% w/v) and the images were acquired with an inverted microscope. The transcriptional profiles from each functional state of lapatinib drug tolerance and resistance development from three parallel BT474 cell plates were surveyed by RNA-sequencing. (B) The number of the genes with a significant change in their expression during the resistance acquisition (see Dataset EV1 for individual gene names). Transcription factor (TF) binding motifs significantly enriched in significantly regulated genes in each transition are indicated. Bolding indicates shared TF binding sites between DTP downregulated and DTEP upregulated genes. (C) Differentially expressed pathways were identified using the R package limma and hallmark gene sets were used for GSVA analysis to reveal hallmarks and signal transduction pathways involved in each step of the resistance acquisition. Red indicates those hallmarks and pathways that are overlapping with processes regulated by DUSP6 depletion in Fig. 3F. Source data are available online for this figure.

## Gene regulatory mechanisms and hallmarks associated with HER2i resistance transitions

To understand gene regulatory mechanisms controlling transitions between the different phases on lapatinib tolerance and resistance development, we predicted the transcription factor binding sites enriched on promoter regions of the differentially regulated genes (Fig. 1B, Dataset EV3). Using FDR < 0.05 as a cut-off, the most highly enriched transcription factor elements in genes down-regulated upon the control-DTP transition were E2F4, FOXM1, FLI1, E2F1, and NFYA (Fig. 1B, Dataset EV3). Strikingly, most of these transcription factor binding sites were enriched also in the genes significantly upregulated in DTEP cells (Fig. 1B in bold, Dataset EV3). This indicates that these transcription factors are inactivated co-ordinately upon the development of the DTP state and reactivated upon regrowth of the DTEPs. E2F4, FOXM1, and E2F1 are all indicated in development and progression of breast cancer. We validated inhibition of selected BT474 cell DTP FOXM1 and E2F target genes across different HER2+ cells treated with lapatinib to reach the DTP state (Fig. EV2). On the other hand, binding sites for transcription factors CBX2, ESR1, TCF12, and ZNF217 were found significantly enriched in the genes down-regulated upon DTP to DTEP transition, whereas STAT1 and STAT2 were enriched among the genes upregulated in LR cells as compared to DTEPs (Fig. 1B, Dataset EV3).

To identify cancer hallmark processes involved in each transition phase, the entire transcriptomics data was re-analyzed by Gene Set Variation Analysis (GSVA). Consistent with recent evidence from other DTP models (Chang et al, 2022; Dhimolea et al, 2021; Kurppa et al, 2020), MYC signaling was inhibited in the lapatinib-treated BT474 DTPs, but reactivated in DTEPs (Fig. 1C, gene sets 3 and 6). Similar gene regulation pattern was observed for E2F1 targets, G2/M checkpoint, mTOR signaling, and androgen response (Fig. 1C, gene set 3), ROS signaling, oxidative phosphorylation, and DNA repair (Fig. 1C, gene set 6). Interestingly, the DTP-DTEP transition was associated also with downregulation of several cancer hallmark gene sets, most apparently seen in set 4 where KRAS signaling, EMT, WNT/ß-catenin, and inflammatory response genes were all suppressed upon proliferation reactivation (Fig. 1C). Additional hallmark gene sets regulated between the lapatinib resistance development transitions are displayed in Appendix Fig. S1B.

Collectively these results demonstrate that unique gene clusters and biological processes are involved in each step of lapatinib resistance development. This bulk RNA-sequencing data provides a rich resource for future studies of gene regulatory mechanisms in HER2i tolerance and resistance development. However, it is clear that future single-cell RNA sequencing studies are needed to understand clonality of gene expression changes induced during these transitions.

## Phosphatase gene expression landscape in DTPs and DTEPs

Recent data indicates that development of cancer therapy resistance is initiated by non-genetic signaling rewiring mediated by post-translational regulation of intracellular signaling pathways (De Conti et al, 2021; Hata et al, 2016; Marine et al, 2020). Protein phosphorylation is the most prevalent post-translational

modification in cancer cells, and cancer cell phosphoproteomes are regulated by kinases and phosphatases. Whereas the role of kinases in non-genetic therapy tolerance development has been extensively studied (Marine et al, 2020), the importance of phosphatases to development of non-genetic kinase inhibitor therapy tolerance have been thus far poorly characterized. Therefore, we focused on the dynamics of phosphatase gene regulation during lapatinib tolerance development. Importantly, there were only four phosphatase genes (CDC25A, CDC25C, DUSP6, and SYNJ1) that were synchronously downregulated in DTP cells but upregulated in DTEPs versus DTPs (Fig. 2A–C, Dataset EV1). We rationalized that these four phosphatases might be particularly relevant for allowing the regrowth of the DTEP population under continuous lapatinib treatment. Even though a large group of other genes were found differentially regulated between the DTEP and LR populations (Fig. 1B, Dataset EV1 and EV2), none of the phosphatase genes were significantly regulated in the DTEP-LR transition (Dataset EV1). This indicates that regulation of phosphatase gene expression could be primarily relevant to the early non-genetic phases of acquired lapatinib resistance.

## Clinical association of DUSP6 with poor prognosis HER2+ breast cancer

Based on the above analysis, CDC25A, CDC25C, DUSP6, and SYNJ1 were the only phosphatases that were significantly regulated during both therapy tolerance transitions (Fig. 2A–C). To evaluate potential clinical relevance of the selected four phosphatases in human breast cancer, we first examined their expression levels across breast cancer subtypes in the METABRIC dataset (Dataref: Pereira et al, 2016). Interestingly, DUSP6 was the only one of these four phosphatases that was selectively overexpressed in the target HER2+ subtype of breast cancers (Fig. 2D). The closest functional orthologue for DUSP6, DUSP1 did not show HER2+ selective overexpression (Appendix Fig. S2A). DUSP6 overexpression in HER2+ breast cancer was confirmed in the TCGA breast invasive carcinoma dataset (Dataref: Cerami et al, 2012; de Bruijn et al, 2023; Gao et al, 2013) (Appendix Fig. S2B). Further, when the 1082 breast invasive carcinoma samples from the TCGA BRCA-dataset (Dataref: Cerami et al, 2012; de Bruijn et al, 2023; Gao et al, 2013) were divided to DUSP6high and DUSP6low groups based on their DUSP6 mRNA expression levels, HER2+ and luminal A subtypes were clearly enriched among the DUSP6high samples (Fig. 2E). HER2 positivity was also found enriched among DUSP6high tumors based on HER2 immunohistochemistry of samples available for staining in the Breast Invasive Carcinoma (TCGA, Firehose legacy) dataset (Dataref: Cerami et al, 2012; de Bruijn et al, 2023; Gao et al, 2013) (Appendix Fig. S2C).

To study the prognostic value of DUSP6 expression in HER2+ breast cancers from the TCGA BRCA-dataset, the patients with high ERBB2 (gene coding for HER2) expression were further divided into DUSP6high and low groups. Comparison of the survival outcomes between the two groups indicate that high DUSP6 expression predicts poor overall and disease-free survival among HER2+ patients (Fig. 2F,G). In multivariable analysis of high ERBB2 expressing tumors, increase in DUSP6 expression and large tumor size (T4) remained significant independent prognostic factors (Appendix Fig. S2D). However, DUSP6 mRNA expression was neither a prognostic factor in Luminal B or Basal subtypes, nor

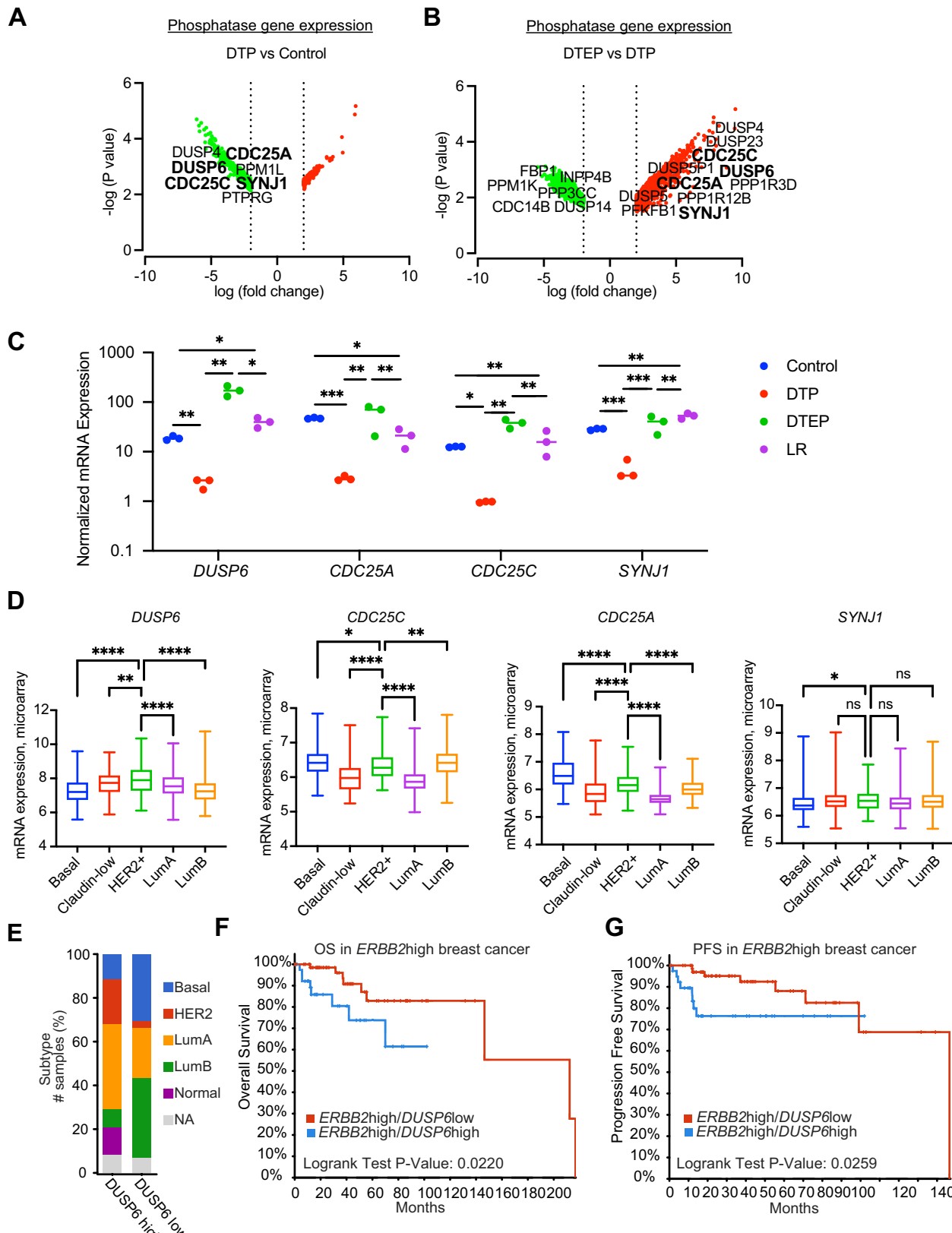

**Figure 2. Clinical association of *DUSP6* overexpression with poor prognosis HER2+ breast cancers.**

(A, B) Volcano plots visualizing differentially expressed genes in (A) Control-DTP and (B) DTP-DTEP transitions. The volcano blots indicate all genes that were significantly regulated during these transitions (|logFC| < 2 and FDR < 0.05), whereas only the phosphatase genes among these are indicated by names. The four phosphatase genes significantly regulated in both transitions (*DUSP6, CDC25A, CDC25C,* and *SYNJ1*) are indicated in bold. Differentially expressed genes were identified using the R package limma (n = 3). (C) Changes in the *DUSP6, CDC25A, CDC25C,* and *SYNJ1* mRNA levels during the acquisition of lapatinib resistance in BT474 cells. Data is based on RNA sequencing analysis (Dataset EV1) and was analyzed by one-way ANOVA followed by Tukey's multiple comparisons test. Statistically significant values of *$p < 0.05$, **$p < 0.01$, and ***$p < 0.001$ were determined (n = 3). (D) Differential expression of *DUSP6, CDC25A, CDC25C,* and *SYNJ1* in different breast cancer subtypes. Data were extracted from the METABRIC dataset and categorized into five molecular subtypes according to the PAM50 gene expression subtype classification (basal, claudin-low, HER2+, Luminal A, and Luminal B). Data were analyzed by one-way ANOVA followed by Tukey's multiple comparisons test and shown as mean ± standard deviation (SD). Statistically significant values of *$p < 0.05$, **$p < 0.01$, and ****$p < 0.0001$ were determined (basal = 209, claudin-low = 218, HER2+ = 224, LumA = 700 and LumB = 475). (E) Breast cancer patients from the TCGA-BRCA dataset were divided into *DUSP6* high (LogFC>1, FDR < 0.05) and low expression (LogFC < −1, FDR < 0.05) groups and the clinical breast cancer subtypes were compared between the two groups. NA; not available. (F, G) Subgroup of 113 patient cases with high tumor *ERBB2* mRNA expression (LogFC>1, FDR < 0.05) were divided into *DUSP6*high and *DUSP6*low groups and their overall survival (OS) (G) (Log-rank Test *p* value = 0.0220) and disease-specific progression-free survival (PFS) (H) (Log-rank Test *p* value = 0.0259) was tested according to *DUSP6* status. Source data are available online for this figure.

across unselected breast cancer patient population (Appendix Fig. S2E), further highlighting the selective connection between DUSP6 and HER2 in breast cancer progression.

Interestingly, luminal B cancers were clearly enriched in *DUSP6*low samples as compared to *DUSP6*high (Fig. 2E). Luminal B are estrogen receptor (ER) positive cancers that often also express HER2. Clinically luminal B cancers are less aggressive than HER2+ cancers. This data may suggest that DUSP6 has a role in suppressing ER positivity, and thereby increasing the relative numbers of HER2+ cancers over luminal B cancers. Although the details of this regulation remain unclear, this hypothesis is supported by strongly decreased expression of ER protein and *ESR1* mRNA, coding for ER, in HER2+ *DUSP6*high samples as compared to HER2+ *DUSP6*low samples (Appendix Fig. S2F,G).

These data demonstrate that DUSP6 is clinically associated with the aggressive HER2+ breast cancer but may also have a broader role in defining breast cancer subtype development.

## DUSP6 promotes the HER2i tolerance and DTP-DTEP transition under continuous lapatinib treatment

Clinical relevance of DUSP6 in HER2+ breast cancers among the four candidate phosphatases differentially regulated at DTP-DTEP transition, together with the feasibility of DUSP6 targeting by small molecules (Kong et al, 2023; Korotchenko et al, 2014; Molina et al, 2009; Vainonen et al, 2021; Zandi et al, 2022), motivated us to select DUSP6 as the phosphatase to be focused in this study. We confirmed differential expression of DUSP6 across different resistance acquisition transitions in BT474 cells. Consistent with the RNA sequencing results (Fig. 2C), DUSP6 protein expression was strongly induced upon the DTP-DTEP transition (Fig. 3A). While the *DUSP6* mRNA levels were diminished in the fully resistant LR clones compared to DTEPs (Fig. 2C), its protein levels remained robustly elevated presumably via post-translational stabilization mechanisms (Fig. 3A). Validating that these effects were not cell line specific, DUSP6 protein was also increased in BT474BrLR cells as compared to the parental cells (Appendix Fig. 3A). We further validated inhibition of *DUSP6* upon DTP phase across independent set of lapatinib treated HER2+ cells including BT474Br, and two cell lines (EFM192A, HCC1419) from (Data ref: Chang et al, 2022) (GSE155342), or NSCLC, melanoma, and colorectal cancer cells de novo treated with various TKIs (Fig. 3B). Therefore, inhibition of *DUSP6* expression appears as a

general mechanism associated with establishment of TKI-induced therapy tolerance. Selected E2F1 and FOXM1 target genes were also confirmed to be downregulated in TKI-treated NSCLC, MM, and CRC DTP cells (Appendix Fig. S3B). Notably, among the transcription factors differentially implicated upon HER2i tolerance development (Fig. 1B), forkhead box transcription factor M1 (FOXM1) and NFYA bind to DUSP6 promoter (Dataset EV4), and coinciding with *DUSP6* expression, *FOXM1* gene expression is downregulated in DTPs and upregulated in DTEPs (Appendix Fig. S3C). Functionally, a small molecule inhibitor of FOXM1 (FDI-6) (Wang et al, 2021) inhibited DUSP6 expression in a time-dependent fashion in BT474 cells (Appendix Fig. S3D,E). Together with its role as breast cancer oncogene involved in therapy resistance (Zhang et al, 2021), these data imply FOXM1 as a viable candidate inducing *DUSP6* expression during the DTP-DTEP transition.

To evaluate whether the hallmark gene sets co-regulated with *DUSP6* during therapy tolerance transitions (Fig. 1C) could be functionally downstream of DUSP6, we performed a RNAseq analysis from *DUSP6*-depleted MDA-MB-453 cells and compared these two gene sets. Notably, there was a marked overlap between hallmark gene sets from these two conditions in which DUSP6 expression was suppressed (Fig. 1C versus Fig. 3C; overlapping gene sets highlighted in red). Especially interesting finding was that the *DUSP6* knockdown cells displayed a gene expression pattern linked to dormant cancer cells such as inhibition of MYC, E2F1 targets, and the PI3K/AKT/mTOR signaling, as well as activation of the interferon response (Chang et al, 2022; Dhimolea et al, 2021; Kurppa et al, 2020; Marine et al, 2020). The finding that siRNA-mediated *DUSP6* depletion (Fig. 3C) recapitulates the gene expression profile in the lapatinib-induced DTP cells (Fig. 1C), clearly indicates that inhibition of DUSP6-driven gene expression programs functionally contribute to the HER2i-elicited growth inhibition.

To functionally validate that increased DUSP6 expression in DTEP cells contributes to their survival, we ectopically over-expressed DUSP6 in BT474 cells, and subjected the cells to treatment with lapatinib, neratinib, afatinib, or tucatinib. Importantly, DUSP6 overexpression was able to dampen both cell viability inhibition, and apoptosis induction, by all four tested HER2is (Figs. 3D,E and EV3A–D). Mechanistically, the anti-apoptotic activity of DUSP6 in lapatinib treated BT474 cells appeared to be independent of ERK MAPK regulation, as overexpression of the KIM mutant of DUSP6 (R64A,R65A),

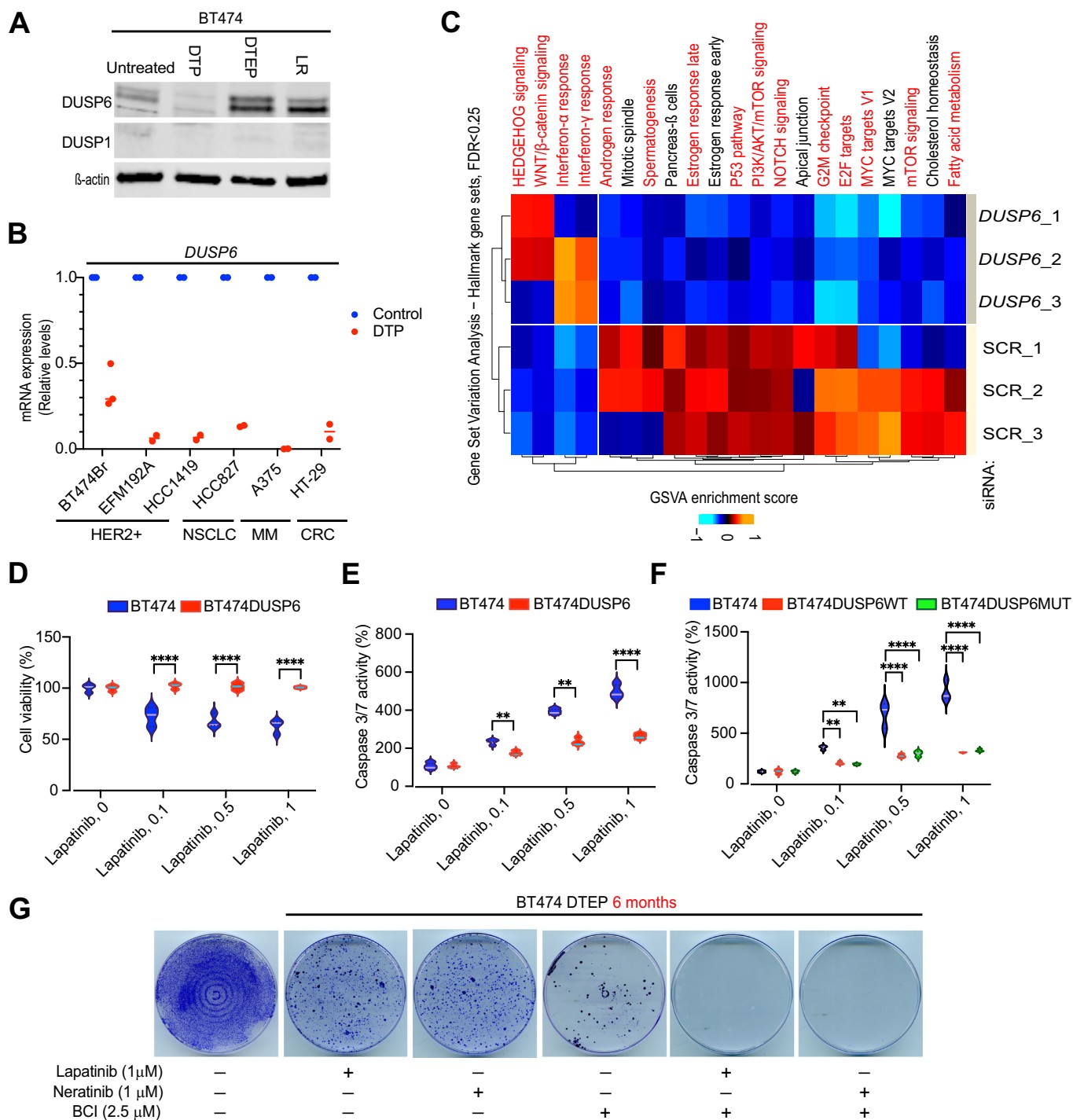

incapable in ERK binding (Nichols et al, 2000), had equally strong impact as the wild-type DUSP6 (Fig. 3F; Appendix Fig. S3F). ERK-independent anti-apoptotic activity for DUSP6 in cancer cells was suggested also recently (Kong et al, 2023). As a complementary pharmacological approach, BT474 cells were treated with lapatinib or neratinib alone, or in combination with small molecule DUSP6 inhibitor BCI for 6 months. BCI is a semi-allosteric inhibitor of both DUSP1 and DUSP6 and several studies have demonstrated that BCI phenocopies genetic DUSP6 inhibition in cancer (Kong

et al, 2023; Ramkissoon et al, 2019; Shojaee et al, 2015). Notably, as compared to the monotherapies, combination with BCI preempted the DTEP development in both lapatinib and neratinib treated cells (Fig. 3G). Strongly indicative of selective drug interaction rather than overall toxicity by BCI, the BCI used at the given concentration for 6 months did not kill all BT474 cells but potently synergized with the HER2is (Fig. 3G). The fact that DUSP6 overexpression inhibited HER2i-elicited cell killing (Fig. 3E), whereas BCI abrogated development of HER2i tolerant cells

**Figure 3. Functional involvement of DUSP6 in HER2i tolerance development.**

(A) Expression of DUSP6 protein in different stages of lapatinib resistance development by Western blot analysis. (B) Relative expression of *DUSP6* mRNA in indicated cell lines either at the untreated control situation, or in the DTP state after following treatments: BT474Br, Lapatinib 1 μM for 9 days; EFM192A and HCC1419, Lapatinib 2.5 μM for 14 days; HCC827 (EGFRmut NSCLC), 1 μM Osimertinib for 10 days; A375(BRAFV600E mutant malignant melanoma (MM)), 1 μM dabrafenib+100 nM trametinib for 10 days; and HT-29 (BRAFV600E mutant colorectal cancer (CRC)), 1 μM dabrafenib+10 μg/ml cetuximab for 10 days. Data for EFM192 and HCC1419 cells was obtained from Dataref: (Chang et al, 2022) (GSE155342), and for other cells by qRT-PCR analysis of the de novo treated samples. Shown is data from two-three repeat samples. (C) Transcriptional profile of MDA-MB-453 cells after *DUSP6* knockdown by 3 different siRNA and compared with 3 different scramble controls, followed by the GSVA analysis of the Hallmark gene sets. The Hallmark gene sets overlapping with the gene sets regulated to same direction in DUSP6 low expressing DTEP cells (Fig. 1C) are indicated with red. (D, E) Ectopic overexpression of DUSP6 in BT474 cells inhibits lapatinib effects on cell viability (D) and apoptosis (E), as measured by WST1 cell viability assay and caspase 3/7 activity, respectively. Data was analyzed by two-way ANOVA followed by Tukey' post hoc test. Statistically significant values of **$p < 0.01$ and ****$p < 0.0001$ were determined. $n = 3$. (F) Ectopic overexpression of either wild-type DUSP6 (DUSPWT) or ERK binding deficient KIM mutant of DUSP6 (DUSP6MUT) in BT474 cells inhibits lapatinib effects on apoptosis. Shown is a result from a representative experiment from three repeats with similar results. Data were analyzed by two-way ANOVA followed by Tukey' post hoc test. Statistically significant values of **$p < 0.01$ and ****$p < 0.0001$ were determined. (G) DUSP6 inhibitor BCI preempts DTEP development in BT474 cells treated with either lapatinib or neratinib for 6 months. The cells were stained and fixed with crystal violet in methanol (0.5% w/v) and the images were acquired with an inverted microscope. Source data are available online for this figure.

(Fig. 3G), strongly support selective effects for BCI on DUSP6. Indeed, the role of DUSP1 as a primary BCI target was overruled by the results that DUSP1 was neither expressed at the protein level in BT474 cells (Fig. 3A), nor its mRNA was found differentially regulated between any of the acquired resistance phases (Fig. 2A,B, Dataset EV1). In harmony, DUSP1 overexpression had clearly weaker activity than DUSP6 overexpression in the rescue experiments (Fig. EV3D). To further validate the selectivity of BCI as a DUSP6 inhibitor, we generated *DUSP6* knockout (*DUSP6*KO) MDA-MB-453 cells by CRISPR/CAS9 (Appendix Fig. S3G). Indeed, three independent single-cell clones of *DUSP6*KO cells were significantly less sensitive to BCI-elicited inhibition of cell viability as compared to the control cells (Appendix Fig. S3H).

Collectively, these results identify *DUSP6* downregulation to be functionally relevant for establishment of the DTP phase, whereas its transcriptional induction contributes to DTP-DTEP transition in lapatinib-treated HER2+ cells.

## DUSP6 targeting kills HER2i resistant breast cancer cells and synergizes with HER2i combination therapies

After discovering the role for DUSP6 in the development of HER2i tolerance, we wanted to address its role in HER2+ breast cancer cells with stable HER2i resistance. To this end, we compared pharmacological DUSP6 targeting against a library of available small molecule modulators of other phosphatases (Lazo et al, 2018; Vainonen et al, 2021). Across either HER2i resistant cells lines, inhibition of cell viability by phosphatase targeting was observed only with DUSP6 inhibitor BCI, its derivative BCI-215, and with FTY-720 that reactivates protein phosphatase 2A (PP2A) by SET inhibition (Saddoughi et al, 2013) (Fig. EV4). The phenocopying results between BCI and BCI-215 is an important additional evidence for the selectivity of BCI type of drugs towards DUSPs, as BCI-215 was demonstrated to activate only the DUSP1 and 6 target MAPKs among the 43 tested kinases (Chan et al, 2020; Kaltenmeier et al, 2017). Based on the response to kinase inhibitors and anti-apoptotic antagonists, the cells that are HER2i resistant, but sensitive to DUSP6 inhibition, are co-dependent on PI3K/AKT, PLK1, and AXL kinase activities, and on the anti-apoptotic proteins IkB, survivin, cIAP and/or XIAP (Fig. EV4). Further corroborating the role for DUSP6 as the anti-apoptotic target in stably HER2i resistance cells, siRNA-mediated depletion of *DUSP6* induced apoptosis in MDA-MB-453 cells (Fig. 4A). This was not seen with *DUSP1* inhibition (Fig. 4A). *DUSP6* depletion also induced

apoptosis in another HER2i resistant HER2+ cell line MDA-MB-361 (Appendix Fig. S4A). Furthermore, independent clones of CRISPR/CAS9 targeted MDA-MB-453 *DUSP6* KO cells showed impaired long-term colony growth potential (Fig. 4B). This was due to loss of DUSP6 expression, as lentiviral re-expression of DUSP6 rescued the phenotype (Appendix Fig. S4B–E).

Notably, the genetic *DUSP6* targeting also sensitized MDA-MB-453 cells to several HER2 targeting approaches. Indeed, whereas the parental MDA-MB-453 cells were resistant to the clinically relevant concentrations of lapatinib, neratinib, and trastuzumab, this resistance was abrogated in the *DUSP6*-siRNA targeted cells (Fig. 4C; Appendix Fig. S4F). These results also validate that DUSP6 inhibition sensitizes to HER2 inhibition regardless of whether small molecule inhibitors or therapeutic antibody (trastuzumab) is used. The impact of genetic *DUSP6* targeting in HER2i sensitization was recapitulated by BCI treatment (Fig. 4D). In addition, *DUSP6* depletion increased sensitivity to the combination of neratinib and capecitabine (Fig. 4E), which is a combination therapy in clinical use (Saura et al, 2020). Furthermore, *DUSP6* depletion enhanced sensitivity to the combination of tucatinib+trastuzumab+capecitabine (Appendix Fig. S4G), which improves progression-free survival and overall survival in patients with HER2+ metastatic breast cancer (Murthy et al, 2020).

Together with DUSP6 overexpression experiments in HER2i sensitive cells (Fig. 3G–I), these data provide strong evidence that DUSP6 contributes to the HER2i resistance both in monotherapy and combination therapy settings.

## DUSP6 inhibition overcomes HER2 inhibitor resistance in vivo

To validate the in vivo relevance of the results, we used both genetic and pharmacological targeting of DUSP6 in HER2i resistant xenograft models. To start with, we evaluated the impact of CRISPR/CAS9-mediated *DUSP6* knockout on the xenograft growth of MDA-MB-453 cells in immunocompromised BALB/cOlaHsd-Foxn1nu mice. Importantly, the two *DUSP6* KO clones showed significant and indistinguishable antitumor effects as compared to the control cells (Fig. 5A). Next, we asked whether pharmacological DUSP6 blockade overcomes HER2i resistance in vivo. For this purpose, we used two different HER2+ cell models, MDA-MB-453 or HCC1954. When xenografts with these two cell lines had reached the approximate size of 100 mm³, the mice were randomized into four treatment groups; vehicle, lapatinib/neratinib

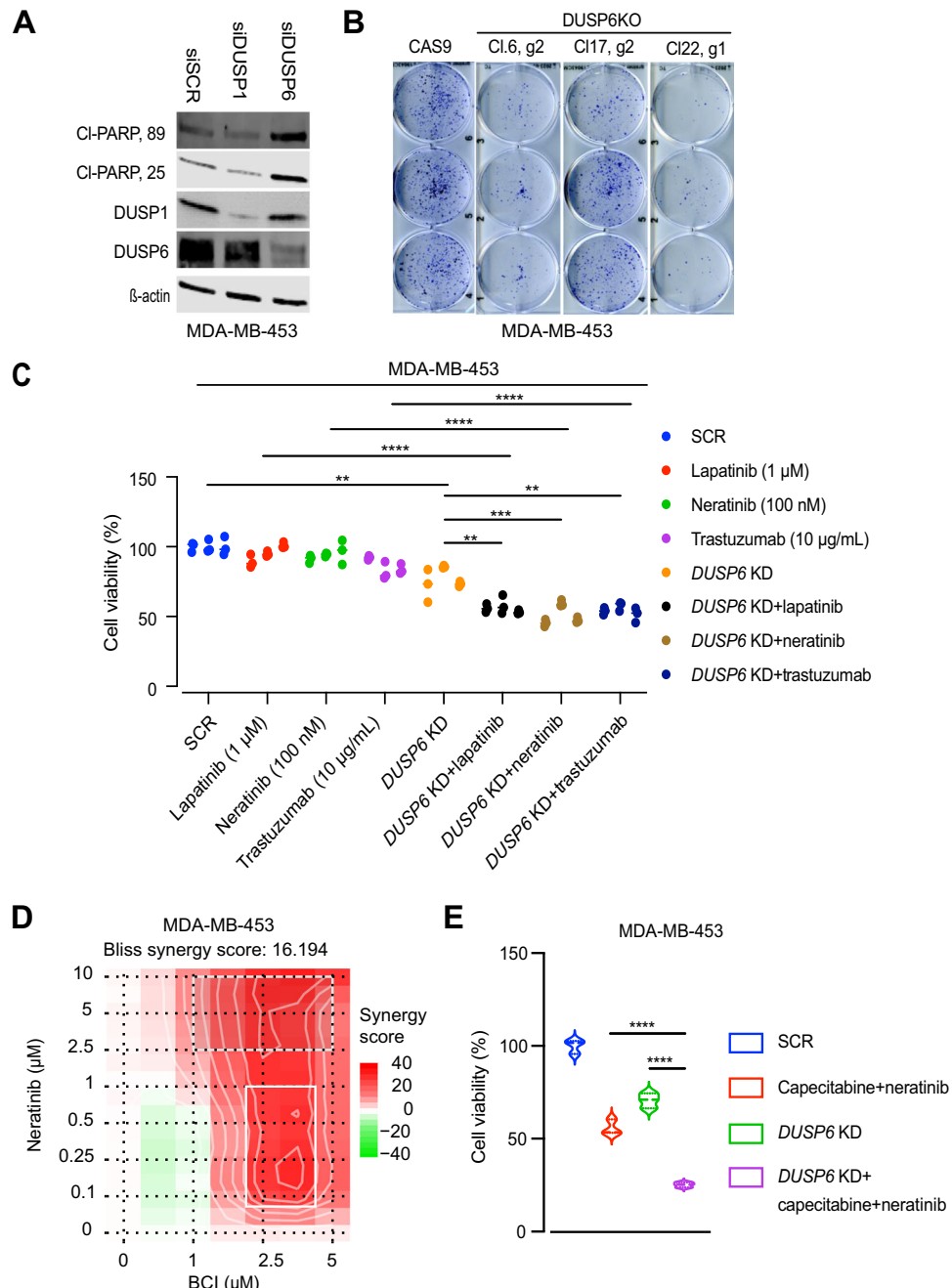

**Figure 4. Both genetic and pharmacological DUSP6 targeting overcomes HER2i resistance.**

(A) RNAi-mediated *DUSP6* knockdown, but not of *DUSP1*, induces apoptotic cell death in MDA-MB-453 cells, as shown by Western blotting for PARP-1 cleavage. (B) CRISPR/CAS9-mediated *DUSP6* knockout hinders the clonogenic growth of MDA-MB-453 cells, as compared to the CAS9 expressing controls. Shown are three independent single-cell clones created with two independent gRNAs (g1 and g2). The cells were seeded at low density and maintained for 10 d. The colonies were stained/fixed with 0.5% crystal violet in methanol and imaged using an inverted microscope. (C) *DUSP6* siRNA knockdown increases sensitivity of HER2i resistant MDA-MB-453 cells to HER2-targeted therapies. Cell viability was measured by WST-1 assay after 48 h of drug treatment. Data were collected from three independent experiments each performed in triplicate and analyzed by one-way ANOVA followed by Tukey's multiple comparisons test. Statistically significant values of $**p < 0.01$, $***p < 0.001$, and $****p < 0.0001$ were determined. (D) A 2D synergy map of neratinib-BCI combination in MDA-MB-453 cells calculated by Bliss SynergyFinder (Ianevski et al, 2017). Higher score (in red) indicates for higher degree of drug synergy. The cultures were treated with increasing concentrations of the compounds for 48 h and cell viability was measured by WST-1 assay. (E) *DUSP6* siRNA knockdown increases sensitivity of HER2i resistant MDA-MB-453 cells to combination with capecitabine and neratinib. Cell viability was measured by WST-1 assay after 48 h of drug treatment. Data were collected from three independent experiments each performed in triplicate and analyzed by one-way ANOVA followed by Tukey's multiple comparisons test. Statistically significant values of $****p < 0.0001$ were determined. Source data are available online for this figure.

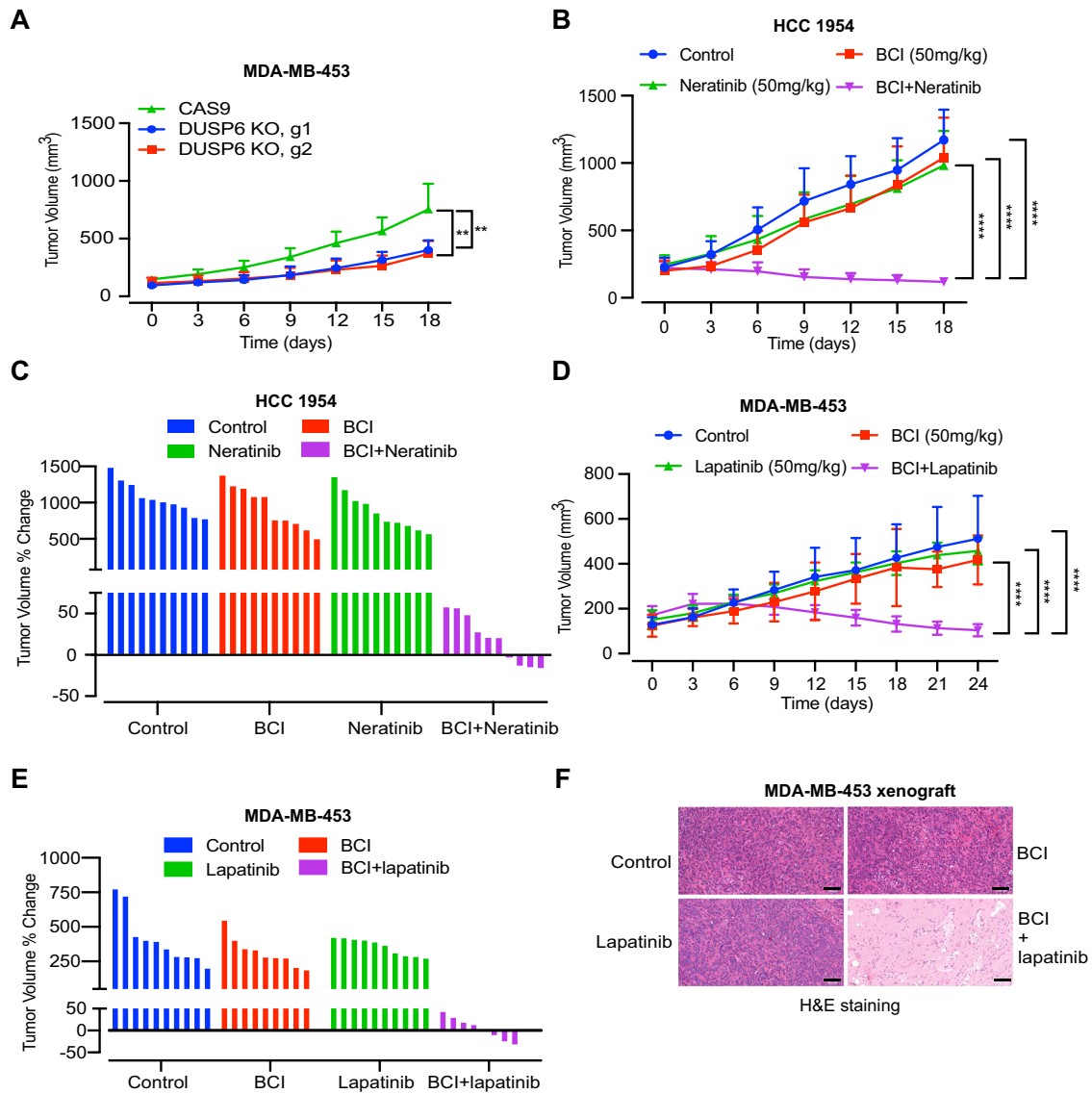

**Figure 5. DUSP6 inhibition overcomes HER2 inhibitor resistance in vivo.**

(A) Subcutaneous xenograft growth of two independent *DUSP6* single cell knockout clones of MDA-MB-453 cells targeted with two different gRNA guides. One-way ANOVA followed by Tukey's multiple comparisons test **$p < 0.01$ comparing each clone to CAS9 expressing control cells. Data are shown as mean ± SD ($n = 5$). (B–E) The effect of BCI in combination with lapatinib or neratinib in two HER2 inhibitor resistant xenograft models: HCC1954 (B, C) or MDA-MB-453 (D, E). Data are shown as mean ± SD ($n = 10$ in each treatment group). Mice with tumor size ~100 mm³ were randomized into the experimental and the control groups and tumor volumes were measured every 3 d. Data were analyzed by one-way ANOVA followed by Tukey's multiple comparisons test. Statistically significant values of ****$p < 0.0001$ were determined. (C) and (E) panels display percentual change in the tumor volume from the start of the therapy as water-fall blots in HCC1954 and MDA-MB453 models, respectively. (F) H&E staining of the representative MDA-MB-453 xenograft tumors from the control, lapatinib, BCI, and lapatinib+BCI groups at day 24. Scale bar 200 μm. Source data are available online for this figure.

(50 mg/kg), BCI (50 mg/kg), and lapatinib/neratinib+BCI. Importantly, validating the in vivo HER2i resistance of both chosen HER2+ cell models, tumors from both cell lines were fully resistant to clinically relevant doses of either lapatinib or neratinib (Fig. 5B–E). Notably, both MDA-MB-453 and HCC1954 tumors also displayed strong resistance to BCI monotherapy (Fig. 5B–E), indicating for tumor microenvironment-mediated impact as compared to the in vitro cultures. However, combination of BCI very efficiently preempted the lapatinib or neratinib resistance phenotype (Fig. 5B–E). Indicating for potential clinical utility,

DUSP6 and HER2-targeted therapies displayed a clear synthetic lethal drug interaction when assessed by a waterfall blot in both cell models (Fig. 5C,E). The dramatic combinatorial activity of DUSP6 and HER2 targeting on cellular viability was also evidenced by lack of cytoplasmic eosin staining of the MDA-MB-453 xenograft tumor after 24 d of treatment (Fig. 5F). Consistent with previous studies with BCI (Kesarwani et al, 2017; Ramkissoon et al, 2019; Shojaee et al, 2015; Wu et al, 2018), we did not observe any apparent signs of toxicity or weight loss (Appendix Fig. S4D) in any treatment groups. In addition, normal fibroblasts displayed several folds

decreased sensitivity to BCI in an in vitro cell viability assay (Appendix Fig. S4E), further indicating for a favorable dose-window between impact of DUSP6 inhibition in HER2i resistance, and its effects to normal cells.

## DUSP6 targeting does not lead to the compensatory HER3 induction characteristic to AKT inhibition

Constitutive activity of PI3K/AKT signaling pathway is strongly associated with HER2i therapy resistance (Berns et al, 2007; Majewski et al, 2015; Nagata et al, 2004). Further, clinical studies of HER2i plus PI3K/AKT inhibitors demonstrated some clinical activity, but did not lead to approval of these combinations (Hudis et al, 2013; Saura et al, 2014). On the other hand, PI3K/AKT activation was one of the DUSP6-driven hallmark gene sets associated with the DTP-DTEP transition (Figs. 1C and 3F). Therefore, it was relevant to compare the quantitative and qualitative differences between DUSP6 blockade versus AKT inhibition in combination with the HER2i in the resistant models. Having demonstrated phenocopying growth effects between genetic *DUSP6* inhibition and BCI, as well as resistance of *DUSP* KO cells to BCI, these experiments were mostly performed by comparing the compounds MK2206 (AKTi) and BCI (DUSP6i) as the alternative pharmacological HER2i combination approaches.

In cell viability assay, both BCI and MK2206 synergized with already low micromolar concentrations of lapatinib and neratinib in MDA-MB-453 and HCC1954 cells, respectively (Fig. 6A,B; Appendix Figs. S4C and S5A). However, as compared to AKT inhibition, DUSP6 targeting had qualitatively superior pro-apoptotic activity. Regardless of efficient inhibition of AKT phosphorylation, MK2206 did not induce apoptosis alone, or in combination with lapatinib (Fig. 6C,D). In contrast, treatment with BCI plus lapatinib triggered apoptosis across all the tested cell models (Fig. 6C,D; Appendix Fig. S5B).

HER3 is a key pro-survival receptor in HER2+ breast cancer cells (Haikala and Janne, 2021). Further, compensatory induction of HER3 is thought to be one of the primary mechanisms behind the resistance to the combination of HER2 inhibition and AKT targeting (Chandarlapaty et al, 2011; Wilson et al, 2012). Accordingly, AKT inhibition resulted in HER3 induction in MDA-MBA-453 cells treated with either MK2206 alone, and with MK2206-lapatinib combination (Fig. 6C,E). Pharmacological inhibition of the AKT upstream kinase PI3K also induced HER3 expression (Appendix Fig. S5C). In contrast, BCI decreased HER2 and HER3 expressions as a monotherapy, and in combination with lapatinib (Fig. 6C,F). Notably, both HER2 and HER3 were downregulated 3 h already after BCI treatment, and this coincided with the induction of ERK phosphorylation, and destabilization of DUSP6, both serving as signs for BCI target engagement (Fig. 6G). Accordingly, siRNA-mediated knockdown of *DUSP6* reduced HER2 and HER3 protein levels in MDA-MB-453, HCC1954, and the HER3+ triple-negative breast cancer cell line MDA-MB-468 (Fig. 6H; Appendix Fig. S5D,E). Indicative of transcriptional regulation, *DUSP6* inhibition decreased *HER2* and *HER3* mRNA expression, but did not impact HER2 and HER3 protein stability (Appendix Fig. S5F,G). Further, indicative of selective effects on HER2 and HER3 among the ERBB receptors, EGFR expression was unaffected by DUSP6 targeting in MDA-MB-468 (Appendix Fig. S5E). The tumor material from the BCI-treated MDA-MB-453 xenograft model was further used for in vivo validation of HER3 as DUSP6

downstream target. Indeed, BCI-treated tumors displayed vastly decreased tumor cell immunopositivity for HER3, and to lesser extent HER2 (Fig. 6I). Finally, clinically high *DUSP6* tumors had significantly higher expression of tyrosine 1298 phosphorylated HER3 (Fig. 6J).

These findings indicate that when HER2i is combined AKTi, the HER2+ cells survive due to compensatory induction of HER3, whereas DUSP6 blockade rather inhibits HER3, and therefore BCI+lapatinib treated cells succumb to apoptosis.

## DUSP6 targeting overcomes NRG-mediated HER2i therapy tolerance

HER3 is activated in tumors by tumor microenvironment derived ligand NRG, and NRG-activated HER3 drives HER2i tolerance (Kodack et al, 2017; Leung et al, 2015). On the other hand, our results indicate that DUSP6 may drive HER2i resistance by promoting the expression of HER3. With this background, we compared the capacity of DUSP6 and AKT inhibition to reverse HER2i tolerance by the NRG/HER3 axis. As expected (Kodack et al, 2017; Leung et al, 2015), NRG treatment induced tolerance to both neratinib and lapatinib across all four tested primarily sensitive cell lines (Fig. 7A; Appendix Fig. S6A). The NRG-mediated rescue from both HER2is was abrogated in a concentration-dependent manner after treatment with BCI, but not with MK2206 (Fig. 7A; Appendix Fig. S6A). The only exception was HCC2218 cells, in which also MK2206 reversed NRG-elicited HER2i tolerance (Fig. 7A; Appendix Fig. S6A). These results were validated at the level of differential apoptosis induction between DUSP6 and AKT targeting. Whereas NRG completely prevented HER2i-elicited induction of apoptosis in BT474 cells, and AKT inhibition could not reverse this, DUSP6 inhibition restored the apoptotic activity (Fig. 7B). On the other hand, RNAi-mediated *HER3* knockdown triggered apoptosis in MDA-MB-453 cells (Appendix Fig. S6B). To functionally prove that inhibition of *HER3* expression mediates BCI effects, we performed a rescue experiment in MDA-MB-453 cells with ectopic stable overexpression of HER3 under heterologous CMV promoter (Appendix Fig. S6C). As compared to control cells, the HER3 overexpressing cells displayed significant resistance to BCI (Fig. 7C).

To link these results back to the development of HER2i tolerance, we found that targeting of DUSP6, but not AKT, reversed the NRG-elicited effects on lapatinib tolerance in the DTP assay (Fig. 7D). This BCI effect was likely due to DUSP6 inhibition, as siRNA-mediated *DUSP6* depletion also abrogated the NRG-induced cell survival in BT474 cells (Appendix Fig. S6D). Importantly, deregulation of NRG-ERBB axis was evidenced also during the DTP-DTEP transition of lapatinib treated cells as, concomitantly with *DUSP6* regulation (Fig. 2C), *NRG3* and *ERBB4* were found upregulated in DTEP versus DTP cells (Appendix Fig. S6E). Further, providing clinical validation for the link between NRG and DUSP6, *NRG* mRNA was significantly overexpressed in breast tumors with high *DUSP6* mRNA expression (Fig. 7E).

Due to brain microenvironment derived NRG, HER3 has a key role in brain metastatic growth of HER2+ breast cancers (Da Silva et al, 2010; Haikala and Janne, 2021). Based on the newly discovered role for DUSP6 in promoting HER3 expression, we investigated contribution of DUSP6 in the brain metastatic outgrowth by using GFP-positive MDA-MB-361 cells as yet additional HER2+ in vivo model. To this end, we quantitated the

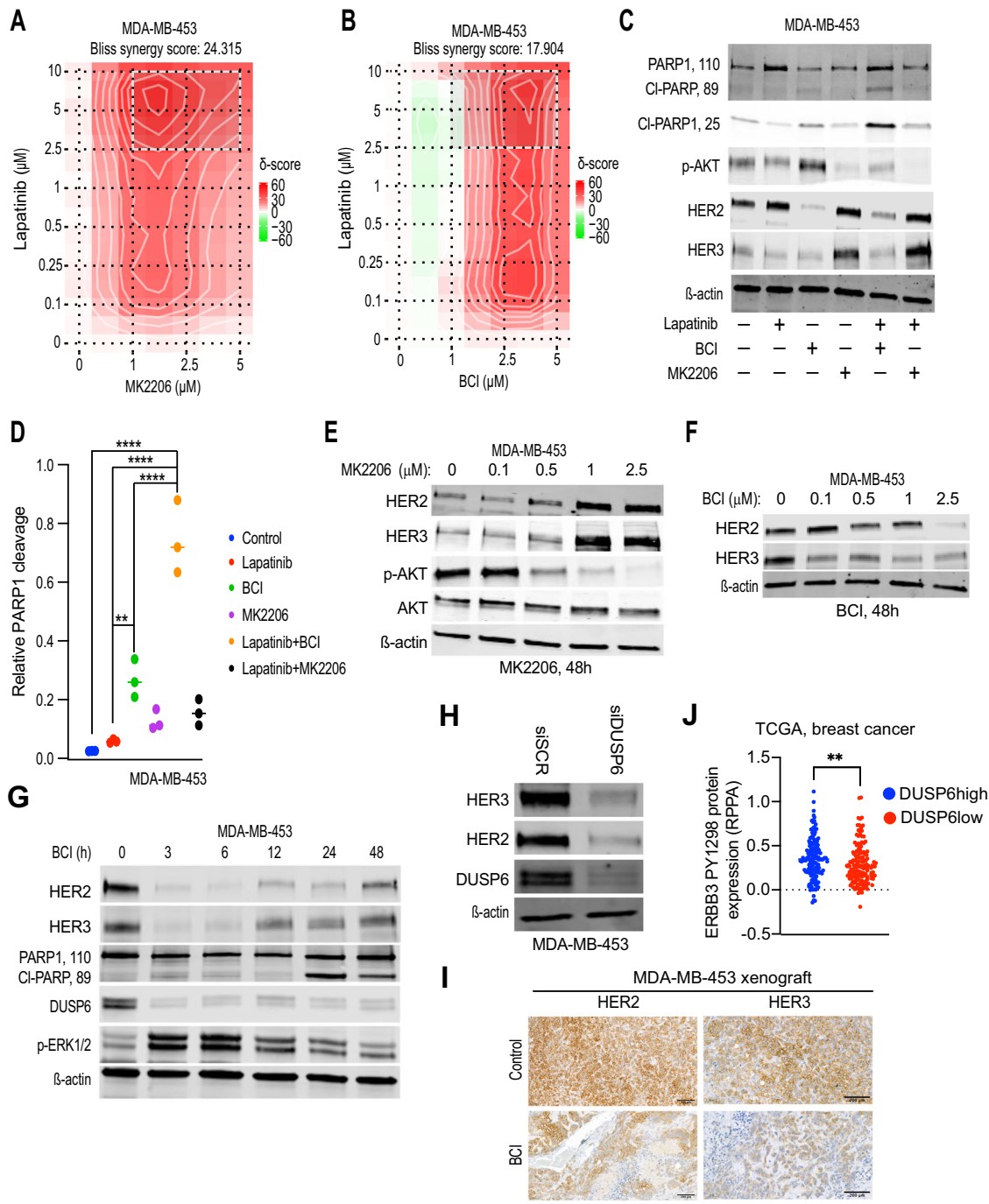

zebrafish intracranial tumor area derived from GFP + MDA-MB-361 cells transfected prior tumor implantation either with control or *DUSP6* targeting siRNA. As shown in Fig. 7F, the zebrafish larva with *DUSP6* targeted cells had significantly smaller tumors 3d after the intracranial injection. Further, mice bearing the intracranial *DUSP6* KO MDA-MB-453 tumors had a significantly longer survival time as compared with the CAS9 control group (Fig. 7G).

Collectively these results demonstrate that DUSP6 inhibition is superior to AKT blockade as a HER2i combination therapy strategy due to its newly discovered capacity to inhibit the NRG-HER3 axis and has potential in inhibiting brain metastatic growth of HER2+ cells.

## A DUSP6-HER3 feed forward loop drives the HER2i tolerance

After identifying previously unrecognized regulation of HER3 by DUSP6 and demonstrating its functional relevance, we asked whether the HER3 reciprocally regulates DUSP6 expression. Indeed, *DUSP6* mRNA expression was potently inhibited already with the smallest tested concentration of neratinib or lapatinib in the HER2i sensitive cells (BT474, BT474Br) (Fig. 8A; Appendix Fig. S7A; in green). However, among the HER2i resistant cell lines, *DUSP6* was not inhibited by HER2is in MDA-MB-361,

**Figure 6. DUSP6 targeting inhibits HER3 expression.**

(A) A 2D synergy map of lapatinib+MK2206(AKTi) or (B) lapatinib+BCI(DUSP6i) combination in MDA-MB-453 cells calculated by the Bliss SynergyFinder. Higher score (in red) indicates for higher degree of drug synergy. The cultures were treated with increasing concentrations of the compounds for 48 h and cell viability was measured by WST-1 assay. (C) Comparison of the effects of lapatinib+BCI and lapatinib+MK2206 on apoptosis induction (PARP-1 cleavage), and HER2 or HER3 protein levels by Western blot analysis. The cells were treated with lapatinib (1 μM), MK2206 (2.5 μM), and BCI (2.5 μM) and their combinations for 48 h. (D) Quantification of PARP1 cleavage from three repeats of (C). Data were analyzed by one-way ANOVA followed by Tukey's multiple comparisons test. Statistically significant values of $**p < 0.01$ and $****p < 0.0001$ were determined. (E, F) The dose-dependent effects of MK2206 (E) and BCI (F) on the expression of HER2 and HER3 protein levels in MDA-MB-453 cells by Western blot analysis after 48 h of treatment. (G) The time-dependent effects of BCI (2.5 μM) on the expression of HER2 and HER3 protein levels and apoptosis induction (PARP cleavage) in MDA-MB-453 cells by Western blot analysis. Increase in phosphorylated ERK (p-ERK1/2) and inhibition of DUSP6 both indicate for early target engagement by BCI. (H) DUSP6 knockdown by siRNA inhibits HER2 and HER3 protein expression in MDA-MB-453 cells. (I) Effects of BCI (50 mg/kg) therapy on HER2 and HER3 protein levels in the MDA-MB-453 xenograft tissue on day 24. Shown is immunohistochemical analysis of HER2 and HER3 from the adjacent paraffin embedded tissue slices from Fig. 5F. Scale bar 200 μm. (J) Breast cancer patients from the TCGA-BRCA dataset were divided into DUSP6 high (LogFC>1, FDR < 0.05) and low expression (LogFC < −1, FDR < 0.05) profiles and expression of phosphorylated HER3 (p-HER3$^{Y1298}$) was compared between the two groups. Data were analyzed by two-tailed t test; $**p < 0.01$ (DUSP6high = 142, DUSP6low = 149). Source data are available online for this figure.

MDA-MB-453, BT474Br-LR, and BT474-LR cells, whereas HCC1954 showed an intermediate phenotype (Fig. 8A; Appendix Fig. S7A; in red). In addition, this regulation was specific to DUSP6, as DUSP1 mRNA expression was not inhibited by HER2 targeting in any of the tested cell modes (Fig. 8A; Appendix Fig. S7A). Clinically DUSP6 and HER3 mRNA expression also correlated in HER2+ cancer samples in the TCGA-BRCA dataset (Cerami et al, 2012) (Appendix Fig. S7B). Differential regulation of DUSP6 expression by HER2i in sensitive (BT474Br, BT474) versus resistant (MDA-MB-361, MDA-MB-453, BT474BrLR) cells was validated at the protein level by western blotting (Figs. 8B,C and EV5). Importantly, RNAi-mediated knockdown of HER3 decreased DUSP6 expression (Fig. 8D). HER3-dependent DUSP6 regulation was further supported by induction of DUSP6 mRNA and protein expression by treatment of serum-starved BT474 or BT474Br cells with the HER3 ligand NRG (Fig. 8E,F; Appendix Fig. S7C,D). Induction of DUSP6 was selective for NRG, since hepatocyte growth factor (HGF), which activates MET RTK, failed to increase DUSP6 in BT474 cells (Appendix Fig. S7C,D). To gain insights into the mechanism by which HER3 regulates DUSP6, the NRG treated cells were co-treated either with AKTi MK2206, or the MEK1/2 inhibitor trametinib. Consistent with a previous report that DUSP6 is a transcriptional ERK1/2 target (Zandi et al, 2022), treatment with trametinib inhibited both ERK phosphorylation and DUSP6 expression in all tested cell lines (Fig. 8E–G; Appendix Fig. S7E), whereas AKT inhibition by MK2206 further increased DUSP6 expression (Fig. 8E–G). Notably, HER2i-elicited inhibition of DUSP6 expression correlated with the capability of HER2i to impact ERK phosphorylation (Figs. 8F,G and EV5A,B).

Collectively these results reveal that inhibition of the newly discovered DUSP6-HER3 feed-forward loop has a major contribution to HER2i therapy response, and that lack of DUSP6 inhibition is a novel HER2i resistance mechanism in acquired resistant cells in which HER2i fails to target MEK activity (Fig. 8H).

## Discussion

Cancer therapy resistance is initiated by non-genetic signaling rewiring resulting in major changes in the epigenetic and transcriptional landscapes of DTPs (Chang et al, 2022; De Conti et al, 2021; Marine et al, 2020; Sharma et al, 2010). It is commonly believed that transition from dormant DTPs to proliferating DTEPs is a pivotal step towards development of the disseminated disease,

and eventually late-stage genetic changes which determine the ultimate therapy resistance (Aguirre-Ghiso, 2021; Marine et al, 2020). Here, we report the first transcriptomic analysis of the DTP-DTEP transition in TKI-treated cancer cells. In addition to importance of the discovery of the role of DUSP6 in DTP-DTEP transition, the transcriptome data will provide a rich resource for future studies to understand different stages of the HER2i resistance development. Based on the fuzzy clustering data (Appendix Fig. S1 and Dataset EV2), the gene clusters 1, 2, and 6, contain genes that are strongly regulated upon the DTP-DTEP transition and are therefore contain many additional genes involved in the evasion of the DTP cells from lapatinib-induced growth inhibition. The longitudinal transcriptomics profiles also allowed us to identify potential master transcription factors involved in the lapatinib resistance development. The functional relevance of the differentially expressed genes associated with each therapy tolerance transition obviously need to be validated in future studies stemming from this novel resource. Further, to understand therapy tolerance development projectories at the single cell level, similar studies are needed by employing single-cell RNA sequencing technologies.

Phosphatases have recently emerged as novel druggable targets for cancer therapy (Lazo et al, 2018; Vainonen et al, 2021; Zandi et al, 2022). However, their contribution to development of non-genetic TKI therapy tolerance is still poorly understood. In the present study, we asked which phosphatases might contribute to the development of HER2i tolerance and resistance and explored their potential role as therapy targets in combination with the HER2i therapies in HER2+ breast cancer. Via transcriptomics analysis of lapatinib tolerance development over six months, we discovered an important role for DUSP6 in the regrowth of HER2i tolerant cells and showed that its genetic and pharmacological blockade potentiates HER2i sensitivity in vitro and in vivo in multiple different HER2+ cell models. Mechanistically, we discovered that transcriptional DUSP6 inhibition is an important contributor to effective HER2i response whereas the lack of HER2i-elicited DUSP6 inhibition in the resistant cells provides a novel explanation for drug resistance in these cells (Fig. 8H). We further discovered DUSP6 as a novel activator of HER3 expression and demonstrated that HER3 inhibition is a prerequisite for the therapeutic impact of DUSP6 inhibition. Inhibition of HER3 by DUSP6 targeting appears also an important mechanistic explanation for the superior apoptotic activity as compared to PI3K/AKT targeting in combination with HER2i. As we demonstrate, the

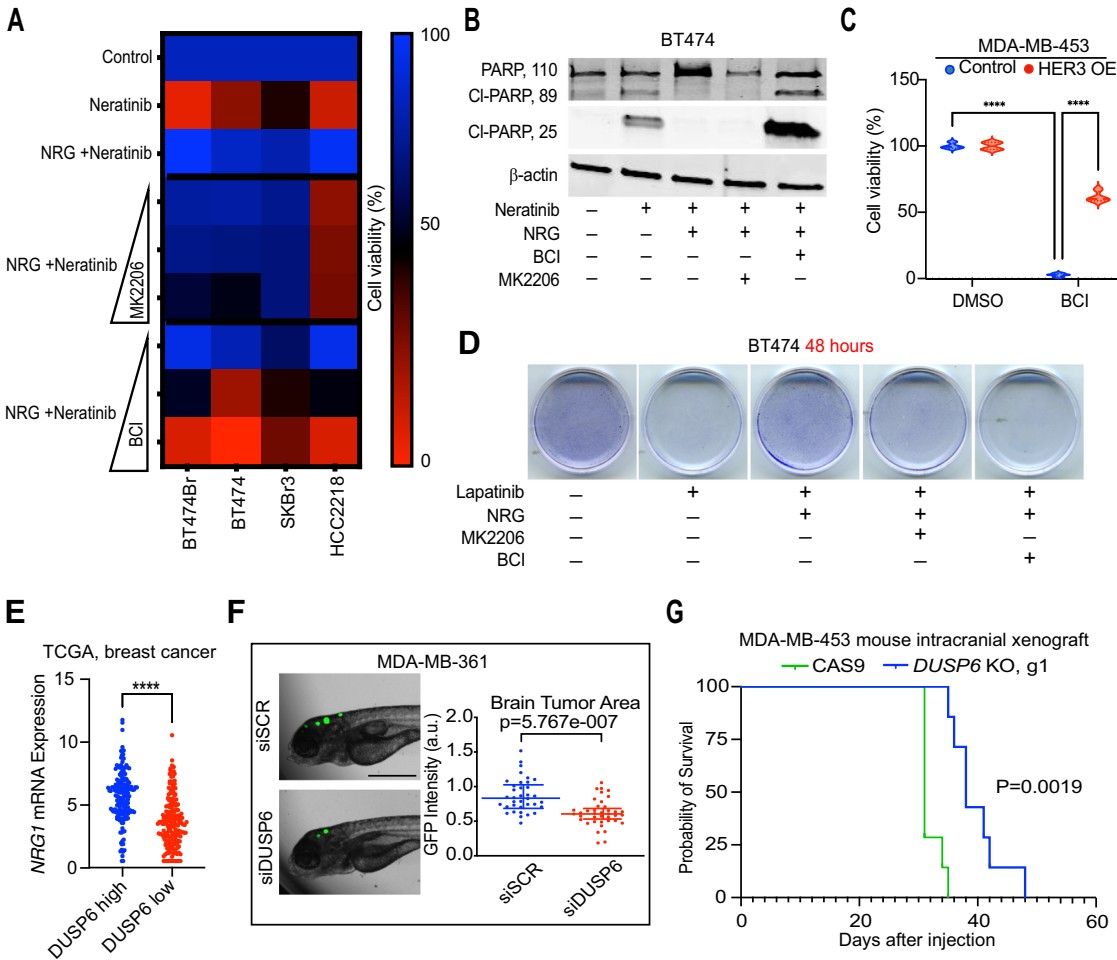

**Figure 7. DUSP6 targeting overcomes neuregulin/HER3-mediated HER2i therapy tolerance.**

(**A**) Comparison of the potential of BCI or MK2206 treatment to overcome neuregulin (NRG)-mediated rescue from the anti-proliferative activity of neratinib. The cells were treated with NRG (10 ng/mL), lapatinib (1 μM), MK2206 (1, 2.5, and 5 μM), and BCI (1, 2.5, and 5 μM) for 48 h and cell viability was measured by WST-1 assay. Data were collected from three independent experiments each performed with three technical repeat samples. (**B**) Comparison of the effects of BCI and MK2206 on NRG-mediated evasion from neratinib-induced apoptotic cell death, as measured by Western blot analysis for cleaved PARP-1. The cells were treated with NRG (10 ng/mL), lapatinib (1 μM), MK2206 (2.5 μM), and BCI (2.5 μM) for 48 h. (**C**) HER3 overexpression rescues MDA-MB-453 cells from BCI-elicited inhibition of cell viability. The control or HER3 overexpressing MDA-MB-453 were treated with BCI (3 μM) for 48 h and cell viability was measured by WST-1 assay. Shown is data from four technical replicate samples from a representative of three experiments with similar results. The data was analyzed by two-way ANOVA + Tukey's post hoc test, ****$p < 0.0001$. (**D**) Comparison of the effects of lapatinib+BCI and lapatinib+MK2206 on NRG-mediated rescue from the anti-growth activity of lapatinib, as shown by crystal violet staining. The cells were treated with NRG (10 ng/mL), lapatinib (1 μM), MK2206 (2.5 μM), and BCI (2.5 μM) for 48 h, stained/fixed with 0.5% crystal violet in methanol and imaged by an inverted microscope (images acquired at ×10 magnification). (**E**) Breast cancer patients from the TCGA-BRCA dataset were divided into *DUSP6* high (LogFC>1, FDR < 0.05) and low expression (LogFC < −1, FDR < 0.05) profiles and the neuregulin (*NRG1*) mRNA levels were compared between the two groups. Data were analyzed by two-tailed t test; ****$p < 0.0001$ (DUSP6high = 167, DUSP6low = 181). (**F**) The effect of *DUSP6* knockdown on the brain metastatic outgrowth of MDA-MB-361 cells in a zebrafish model. GFP-positive MDA-MB-361 cells transfected either with control scrambled siRNA or DUSP6 siRNA were injected into zebrafish embryo brain and the GFP intensity was measured 3 days after by microscopy. Data were analyzed by two-tailed *t* test; ****$p < 0.0001$. Scale bar 100 μm (siSCR = 37, siDUSP6 = 43). (**G**) Improved overall survival of mice with intracranially injected DUSP6 KO MDA-MB-453 cells as compared to the CAS9 positive control cell injected mice. Survival data were analyzed by log-rank Mantel–Cox test, **$p < 0.01$. Source data are available online for this figure.

therapeutic impact of DUSP6 targeting is not compromised by the NRG-mediated feedback to HER3 as is the case with AKT targeting and thereby our data identify DUSP6 targeting as a novel approach to target NRG-HER3 axis to overcome HER2i resistance. These are important advantages favoring DUSP6 blockade versus AKT inhibition as a future strategy for treatment of HER2+ breast cancer. A particularly important translational finding in our study is that DUSP6 defines therapy sensitivity of the lapatinib-resistant cells in response to several ERBB TKIs with different specificities,

and to their chemotherapy combinations. This is consistent with the proposed HER3-DUSP6 positive feedback mechanism, as HER3 or NRG overexpression confer resistance to several cancer drugs (Erjala et al, 2006; Haikala and Janne, 2021; Knuefermann et al, 2003; Recondo et al, 2020; Wilson et al, 2012; Yonesaka et al, 2011). Thereby identification of DUSP6 targeting as a novel approach for HER3 inhibition may have broad ramifications in different combination therapy settings across different cancer types. HER2 and 3 play also essential roles in the development of brain

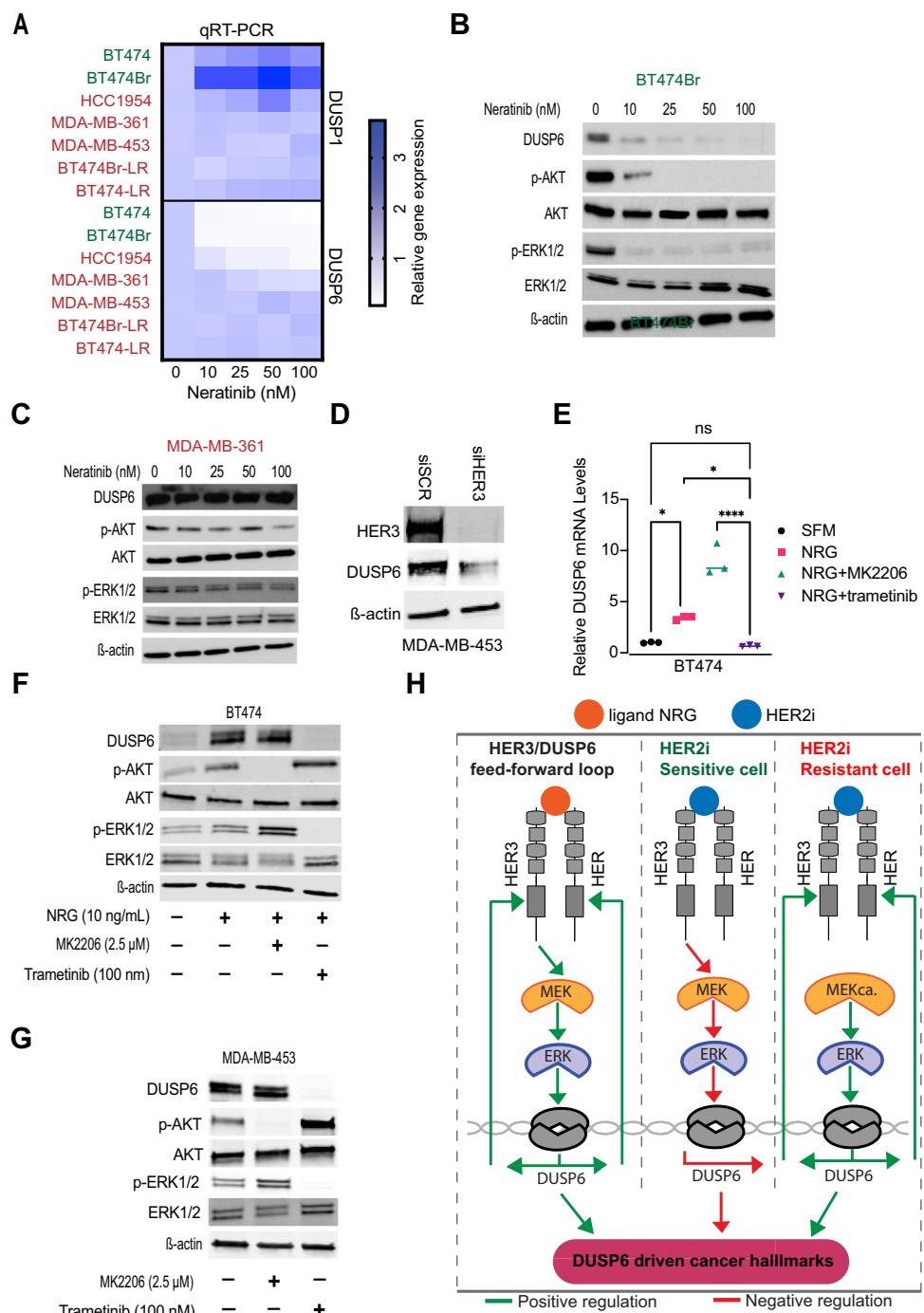

metastasis which causes death of more than 30% of stage IV HER2+ breast cancer patients (Berghoff et al, 2014; Da Silva et al, 2010; Fecci et al, 2019). To this end, DUSP6-mediated regulation of HER2 and 3 suggested that its inhibition may retard development of brain metastasis. Indeed, our findings directly support these conclusions as *DUSP6* knockout in tumor cells improves survival of mice with intracranial HER2+ tumors and the outgrowth of HER2+ breast tumor cells in a zebrafish intracranial model. Linking our results further to future therapy of HER2+ brain metastasis, we showed that *DUSP6* knockdown increases sensitivity

to tucatinib+trastuzumab+capecitabine combination regimen, which show significant clinical activity in HER2+ breast cancer patients with brain metastasis (Murthy et al, 2020).

While demonstrated here for the first time in the context of HER2i therapy resistance, genetic DUSP6 inhibition has recently been shown to inhibit malignant phenotypes in other cancer types (Shojaee et al, 2015; Wu et al, 2018; Zandi et al, 2022). Consistent with these reports, genetic inhibition of *DUSP6* in our study resulted in significant inhibition of HER2+ breast cancer cell viability, HER2i resistance, colony forming potential, and in vivo

◄ **Figure 8.   Feed-forward loop between NRG/HER3 and DUSP6 determines HER2i sensitivity.**

(A) *DUSP1* and *DUSP6* mRNA levels were determined by qRT-PCR analysis after treatment with increasing concentrations of neratinib for 48 h in indicated cell lines. Red denotes for HER2i resistant cell lines and green HER2i sensitive cells. Data were collected from three independent experiments each performed in triplicate. (B, C) Comparison of the effect of neratinib treatment (48 h) on DUSP6, p-AKT, and p-ERK1/2 between HER2i sensitive BT474Br (B) and HER2i resistant MDA-MB-361 (C) cells, respectively. (D) The effect of siRNA-mediated *HER3* knockdown on DUSP6 expression in MDA-MB-453 cells by Western blot analysis. (E) NRG-mediated induction of *DUSP6* mRNA via MEK activation as measured by qRT-PCR analysis after treatment with NRG (10 ng/mL), MK2206(AKTi) (2.5 μM), and trametinib(MEKi) (100 nM) for 48 h. Data were analyzed by one-way ANOVA followed by Tukey's multiple comparisons test. Statistically significant values of *$p < 0.05$ and ****$p < 0.0001$ were determined ($n = 3$). (F) The effect of NRG on DUSP6 protein expression BT474 cells via MEK activation. The cells were serum-starved for 24 h, followed by treatment with NRG (10 ng/mL), MK2206 (2.5 μM) and trametinib (100 nM) for 48 h. (G) Inhibition of DUSP6 expression in HER2i resistant MDA-MB-453 cells by MEKi Trametinib. The cells were treated with MK2206 (2.5 μM) or trametinib (100 nM) for 48 h. (H) A schematic illustration of the discovered HER3/DUSP6 feed forward loop in HER2+ breast cancer cells. NRG binding to HER3 induces MEK/ERK-mediated *DUSP6* expression which feeds back to increased HER2 and HER3 expression (left panel). In HER2i sensitive cells (middle panel) inhibition of HER3 results in *DUSP6* inhibition and loss of DUSP6 driven cancer hallmarks. In HER2i-resistant cells (right panel), MEK is not inhibited by HER2i but its constitutive activity (MEKca.) drives DUSP6-HER2/3 positive feed-back loop resulting in HER3-mediated multitherapy resistance and cancer progression. Source data are available online for this figure.

tumor growth. Mechanistically, DUSP6 inhibition converted cytostatic response to HER2i to an apoptotic response, which is considered as a paramount for cancer therapy strategies aiming for cancer cure. On the other hand, overexpression of DUSP6 converted HER2i sensitive cells to resistant in both cell viability and apoptosis assays. Importantly, all our main conclusions remained valid regardless of whether DUSP6 was inhibited either by siRNA, CRISPR/CAS9, or by pharmacological inhibitors BCI and BCI-215. We further demonstrated nearly immediate target engagement, as well as resistance of three independent *DUSP6* knock-out clones to BCI-elicited cell viability inhibition. Therefore, together with recent results from the others (Kong et al, 2023; Shojaee et al, 2015), we are confident that the results achieved by BCI and BCI-215 are dependent on their inhibitory effects on DUSP6 despite a recent report questioned their selectivity using other cellular system (Thompson et al, 2022).

The results identified DUSP6 as a novel combination therapy target with existing clinical HER2 targeting strategies including both small molecule and antibody-mediated HER2 inhibition and chemotherapy combinations. In that regard, our xenograft results demonstrate significant potential for pharmacological DUSP6 inhibition in overcoming HER2i resistance in vivo by using two different HER2i resistant cell lines, and two different HER2is. Whereas neither HER2 nor DUSP6 inhibition alone did not have therapeutic effect on tumors, the combination showed very potent synthetic lethal phenotype further validating our in vitro results. The lack of monotherapy effect of BCI, combined with clear evidence for downstream target mechanism engagement in vivo also alleviates concerns about overall cellular toxicity behind the BCI-elicited HER2i synergy. This is consistent with previous findings demonstrating antitumor effects with BCI and BCI-215 without obvious systemic toxicity in in vivo models of gastric cancer, leukemia, and malignant peripheral nerve sheath tumor (Kesarwani et al, 2017; Ramkissoon et al, 2019; Shojaee et al, 2015; Wu et al, 2018).

In summary, we provide first transcriptional map of DTP-DTEP transition under TKI tolerance development in cancer. The results specifically identify DUSP6 targeting as a novel approach to target HER3-mediated ERBB TKI resistance. Ultimately, the work provides proof-of-principle evidence to encourage development of next-generation DUSP6 inhibitors (with brain penetrance) to test the clinical relevance of the presented therapy scenarios.

## Methods

### Reagents

Tissue culture reagents including regular RPMI, DMEM, RPMI, and FBS were purchased from Sigma. The recombinant NRG, HGF, and β-estradiol (E2) were from Peprotech. BCI was purchased from Axon Medchem. BCI-215 was provided by Dr. Andreas Vogt, University of Pittsburgh Drug Discovery Institute, Pittsburgh, PA, USA. The HER2 inhibitors and the other compounds used in the drug screening were purchased from Adooq Bioscience. All the agents were dissolved in DMSO and the final concentration of DMSO did not exceed 0.1% [v/v] in all the treatments.

### Cloning and plasmids

pBABE-puro-gateway-ERBB2 was a gift from Matthew Meyerson (Addgene plasmid No. 40978; http://n2t.net/addgene:40978; RRI-D:Addgene_40978). ERBB3 wild-type was cloned from pBABE-puro-gateway-ERBB3 (Koivu et al, 2021) into pLenti CMV Puro DEST (w118-1), a gift from Eric Campeau and Paul Kaufman (Addgene plasmid # 17452; http://n2t.net/addgene:17452; RRI-D:Addgene_17452) through Gateway cloning (Chakroborty et al, 2019) to create pLenti-CMV-Puro-ERBB3. The DUSP6 plasmid was purchased from Addgene (#27975) and the Kinase interaction motif (KIM) mutants were prepared by site directed mutagenesis (GenScript Inc) generating R64A, R65A double mutant defective in ERK binding as described (Nichols et al, 2000).

### Generation of stable lines

To overexpress wild-type ERBB2, pBABE-puro-gateway-ERBB2 was transfected (using Fugene6 transfection reagent; Promega Catalog # E2692) into amphotropic Phoenix HEK293T cells (a gift from Dr. Garry Nolan) to generate retroviruses, which were used to transduce MDA-MB-453, as described previously (Chakroborty et al, 2019). pLenti-CMV-Puro-ERBB3 was co-transfected with virus-packaging plasmids pMLDg/pRRE (addgene #12251), pMD2.G (addgene #12259), and pRSV-Rev (addgene #12253) into HEK293T cells using Fugene6 transfection reagent to produce lentiviruses. The lentivirus-containing supernatant was used to transduce MDA-MB-453 to over-express wild-type ERBB3. After

viral transduction cells were treated with 1 μg/mL puromycin (Gibco) for 48 h to select the cells with stable expression of the respective introduced transgenes.

For DUSP1 and 6 overexpression, lentiviral particles containing full length of either DUSP1 (Genecopoeia), DUSP6 (Addgene #27975), or control empty (Genecopoeia) vector were generated in HEK293FT packaging cell line (complete medium: high glucose DMEM, 10% FBS, 0.1 mM NEAA, 1 mM MEM Sodium Pyruvate, 6 mM L-Glutamine, 1% Pen/Strep and 0.5 mg/ml Geneticin) by transient transfection of transfer vector 2nd generation packaging plasmid-psPAX2 (Addgene #12259) and envelope vector-pMD2 (Addgene #12260) with the ratio (7:2:1) using calcium-phosphate precipitation method. Seventy-two hours post-transfection medium-containing viral vectors was collected, concentrated for 2 h by ultracentrifugation in swing-out rotor SW-32Ti (Beckman Coulter), $26,000 \times g$, resuspended in residual medium and flash-frozen in liquid nitrogen. Functional titer $\sim 1 \times 10^8$ was measured in HEK293FT cells and FACS (BD LSRFortessa, Becton Dickinson). To obtain stable overexpression of DUSP1, DUSP6, or double DUSP1 + 6 population on day zero, $8 \times 10^4$ cells were seeded in a 24-well plate. 24 h later, the cells were transduced with MOI 60 of lentiviral stocks in a low volume of full media. Medium-containing viral particles was removed 16 h later. Cells expressing DUSP1 and GFP indicative of lentiviral integration were collected by fluorescence assisted cell sorting (BD FACSaria II cell sorter, Becton Dickinson). FACS gating was set at 10% top high fluorescence signal. DUSP6 transduced BT474 cells were selected with 3 μg/mL of puromycin. DUSP1 and 6 expressing cells were obtained by sequential transduction (MOI 2, 6, 10), puromycin selection and later GFP fluorescence assisted cell sorting (BD FACSaria II cell sorter, Becton Dickinson). The levels of protein were confirmed by Western blot analysis.

For stable overexpression of DUSP6 and DUSP6 KIM mutant, DUSP6 (Addgene #27975) and DUSP6$^{R64-65A}$ lentiviral vectors were co-transfected with virus-packaging plasmids pMLDg/pRRE (Addgene #12251), pMD2.G (Addgene #12259), and pRSV-Rev (Addgene #12253) into HEK293T cells using Fugene6 transfection reagent to produce lentiviruses. Forty-eight hours after transfection viral supernatants were collected, filtered, and added to pre-plated BT474 cells along with 10 μg/mL polybrene. Transduced cells were selected with 2 μg/mL puromycin, and the protein levels were confirmed by Western blot analysis.

To rescue the DUSP6 KO MDA-MB-453 clones, the pLEX307-hygro-DUSP6 lentiviral vector was co-transfected with virus-packaging plasmids pMLDg/pRRE (Addgene #12251), pMD2.G (Addgene #12259), and pRSV-Rev (Addgene #12253) into HEK293T cells using FuGENE6 transfection reagent. Forty-eight hours after transfection viral supernatants were collected, filtered, and added to MDA-MB-453 DUSP6 KO cells (g1, clone 22) along with 10 μg/mL polybrene. Transduced cells were selected with hygromycin (250 μg/ml). The MDA-MB-453 CAS9 control, DUSP6 KO, and DUSP6 transduced KO cells were seeded at low density and maintained for 10 d. The colonies were stained with 0.5% crystal violet in methanol and imaged using an Epson scanner. Colony quantification was done by using the ImageJ ColonyArea plugin.

## FACS

MDA-MB-453 cells transduced with lentiviruses encoding wild-type ERBB3 were washed with azide-free PBS, trypsinized, and suspended in ice-cold sorting buffer (PBS + 1% Goat serum, Life Technologies catalog # PCN5000). The cells were incubated with anti-ERBB3 (MAB3481, R&D Systems) for 1 h on ice, and with anti-mouse Alexa Fluor 405 (A-31553, Invitrogen) for 30 min on ice in a dark environment. Single cell suspension was analyzed and sorted on Sony SH800 Cell Sorter to select cell-pools with high surface ERBB3 expression.

## CRISPR/CAS9 knockout system

A two-component CRISPR system was used to generate DUSP6 KO cells (Adli, 2018). DUSP6 sgRNAs (seq#1- CATCGAGTCGGC-CATCAACG, seq#2-GACTGGAACGAGAATACGGG, seq#3-CCATGATAGATACGCTCAGA) were selected using DeskGEN platform and cloned according to F. Zhang lab protocol. Separate lentivectors containing spCas9 (lentiCas9-Blast a gift from Feng Zhang (Addgene plasmid # 52962) and sgRNA (lentiGuide-Puro a gift from Feng Zhang (Addgene plasmid # 52963) were produced in HEK293FT packaging cell line by transient cotransfection. Shortly, 40–70% confluent HEK293FT cells were used for transfections with 14 μg of transfer vector, 4 μg of packaging vector psPAX2 (gift from Didier Trono (Addgene plasmid # 12260), 2 μg envelope vector pMD2.G (gift from Didier Trono (Addgene plasmid # 12259) mixed in 0.45 mL water, 2.5 M CaCl$_2$, and 2x HeBS (274 mM NaCl, 10 mM KCl, 1.4 mM Na$_2$HPO$_4$, 15 mM D-glucose, 42 mM Hepes, pH 7.06) per 10 cm dish. Before adding to the cells, the DNA-HeBS mix was incubated for 30 min at room temperature. After overnight incubation medium with DNA precipitate was gently removed from the cells and replaced with a full fresh medium. Media containing viral particles was collected after 72 h, spun at 300 rpm for 5 min at room temperature to remove cell debris, filtered through 0.45 μm PES filter, and concentrated by ultracentrifugation for 2 h at 25,000 rpm, 4 °C (Beckman Coulter). The pellet containing lentiviral particles was suspended in the residual medium, incubated for ~2 h in +4 °C with occasional mild vortex, aliquoted, snap-frozen, and stored in −70 °C. P24 ELISA measured physical lentiviral titer with a serial dilution of virus stock according to manufacturer protocol.

To generate DUSP6KO clones in MDA-MB-453, 1e + 05 cells were seeded on a 24-well plate. The next day the cells were transduced with Lenti-Cas9 (MOI 1, 5, 10), and 72 h later, 8 μg/mL of Blasticidin was applied to select only Cas9 expressing cells. Cells transduced with the smallest number of Lenti-Cas9 particles that survived after the parallel control well was cleared proceeded to the next step. In the second stage, the mixed pool of stably expressing Cas9 cells was transduced with Lenti sgRNA vectors (MOI 5,10,15,20), and 72 h later, 1 μg/uL Puromycin was applied on the cells to select double-positive Cas9+/Lenti sgRNA+ cells. Based on Western blot results, cell populations showing the highest reduction in DUSP6 protein levels were single sorted (Sony SH800 cell sorter, Sony Biotechnology Inc) and re-grown into a clonal cell population. On average, about 20 clones per sgRNA population were screened using Western blotting and qPCR. Sanger sequencing was used to confirm full knockout status.

## Cell Culture and Transfections

All cell lines were purchased from the American Type Culture Collection (ATCC) and were maintained at 37 °C and 5% CO$_2$ in a

humidified incubator and cultured according to the ATCC recommendations. Cell line authentication was performed by STR profiling and using Cell ID™ system (Promega). The cultures were routinely tested for mycoplasma contamination. The information about siRNAs for *DUSP1*, *DUSP6*, *HER2*, *HER3*, *AKT1* and negative control siRNAs are in Table EV1. Transient transfections were performed with lipofectamine RNAiMAX reagent (Thermo-Fisher) according to the manufacturer's instructions.

## Cell growth assays

The cells were seeded at a density of $2 \times 10^3$ into 96-well plates and treated with increasing concentrations of the drugs for 48 h. Cell viability was determined using WST-1 assay (Sigma). Vehicle-treated cells were used as the control group. For the clonogenic survival assay, cells were seeded in 6-well plates at a density of 1000 cells/well. After 24 h, the media was changed, and the cells were maintained for another 10 d. The resulting colonies were stained/fixed with 0.5% crystal violet imaged using an inverted microscope.

To yield DTPs in NSCLC, melanoma, and colorectal cancer cell lines, the following treatments were carried out: NSCLC cell line HCC827, melanoma cell line A375, and colorectal cancer HT-29 cell line were cultured under standard conditions. The following treatments were applied for 10 d: HCC827, 1 μM osimertinib; A375, 1 μM dabrafenib+100 nM trametinib; HT-29, 1 μM dabrafenib +10 μg/ml cetuximab.

## Drug combination analysis

To explore the efficacy of drug combinations, growth inhibition was determined by WST-1 assay and the results were evaluated by Bliss SynergyFinder (Ianevski et al, 2017). Visualization of synergy scores is depicted as a synergy map. An average synergy score of 0 is considered additive, less than 0 as antagonistic and over 0 as a synergistic outcome.

## Caspase 3/7 activity assay

The Caspase-Glo® 3/7 Assay System (Promega), a luminescence-based assay for detection of active caspase-3 and 7, was used to quantitatively determine apoptotic cell death. Following caspase cleavage of the proluciferin DEVD substrate, a substrate for luciferase is released and results in luciferase reaction and production of light.

## Western blot analysis

The cells were lysed for 30 min in ice-cold RIPA buffer (50 mM Tris-HCl, pH 8.0, 150 mM NaCl, 1.0% NP-40, 0.5% sodium deoxycholate and 0.1% SDS) containing protease and phosphatase inhibitors (ThermoFisher). The assay was performed with primary and Licor secondary antibodies (Licor Biosciences). β-actin was used as the loading control. The list of antibodies is in Table EV2.

## Analysis of gene expression by quantitative reverse transcription-PCR

Quantitative reverse transcription-PCR (qRT-PCR) analysis was done on a QuantStudio Real-Time PCR instrument (ThermoFisher) using PowerUp™ SYBR® Green Master Mix (ThermoFisher). The primer sequences are listed in Table EV3. The target gene expression levels were normalized to beta-2-microglobulin (*B2M*) levels. For calculations, $2^{-\Delta\Delta CT}$ formula was used, with $\Delta\Delta CT = (CT_{Target} - CT_{B2M})$ experimental sample $- (CT_{Target} - CT_{B2M})$ control samples, where CT is the cycle threshold.

## Animal studies

The animal experiments were performed according to the Animal Experiment Board in Finland (ELLA) for the care and use of animals under the licenses 4161/04.10.07/2015 and 9241/2018. The animals were kept under pathogen-free conditions in individually ventilated cages in an animal care facility. Mice were kept on a 12 h light/dark cycle with access to autoclaved water and irradiated chow *ad libitum* and were allowed to adapt to the facility for 1 week before starting the experiments. For the subcutaneous experiments, MDA-MB-453 ($5 \times 10^6$) and HCC1954 ($3 \times 10^6$) cells were injected into the right flank of 6- to 8-week-old BALB/cOlaHsd-Foxn1nu mice (Envigo, France). Mice with tumor size ~100 mm³ were randomized into experimental and control groups. Tumor dimensions were measured with Vernier calipers and tumor volume were calculated as 1/2 larger diameter x (smaller diameter)². For the intracranial model, $1 \times 10^5$ cells in 5 μL of PBS were inoculated into the brain of anaesthetized mice. The mice were then imaged with bioluminescence and based on the bioluminescence signal were randomized to experimental and control groups, as descried earlier (Merisaari et al, 2020). Mice were euthanized when they became moribund when they reached defined study end points.

## Zebrafish studies

Zebrafish embryo xenograft studies were performed under the license ESAVI/9339/04.10.07/2016 (National Animal Experimentation Board, Regional State Administrative Agency for Southern Finland). Briefly, the *DUSP6* knockdown GFP-MDA-MB-361 cells were injected into the brain of zebrafish embryos from the dorsal side. One day after injection (1 dpi), successfully transplanted embryos were placed in CellView glass bottom 96-well plate (1 embryo/well) and embryos were incubated in E3 + PTU at 33 °C. The xenografted embryos were imaged using a Nikon Eclipse Ti2 fluorescence microscope and a 2x Nikon Plan-Apochromat (NA 0.06) objective. Each embryo was imaged at 1 dpi and 4 dpi using brightfield illumination and a GFP fluorescence filter set (excitation with 470 nm LED). Each image was inspected manually to filter out severely malformed, dead or out of focus embryos. The tumor area was measured using ImageJ (NIH). The fold change in tumor size was calculated as follows: GFP intensity (4 dpi)/GFP intensity (1 dpi).

## RNA-sequencing

RNA-sequencing was conducted at the Finnish Functional Genomic Center, The University of Turku, Finland. RNA was harvested using the NucleoSpin RNA purification kit (Macherey-Nagel), followed by treatment with DNase to remove genomic DNA. RNA (300 ng) was reverse transcribed using the Illumina TruSeq Stranded Total mRNA kit. The quality of the samples was

ensured using Agilent Bioanalyzer 2100 or Advanced Analytical Fragment Analyzer. Sample concentration was measured with Qubit® Fluorometric Quantitation (Life Technologies) and/or KAPA Library Quantification kit for Illumina platform, KAPA Biosystems. Sequencing run was performed using the Illumina NovaSeq 6000 instrument. Genomic alignment was performed using Rsubread v. 2.0.0 and the reads were mapped to the human reference genome hg38. Aligned reads were assigned to RefSeq gene models using the same R package with its default settings.

Differentially expressed genes and pathways were identified using the R package limma. Latest hallmark gene sets were downloaded from the Molecular Signatures Database version 7.4 (http://www.gsea-msigdb.org/gsea/msigdb/index.jsp) and used within Gene Set Variation Analysis (GSVA) allowing pathway enrichment estimates for each sample (Hanzelmann et al, 2013). Data was transformed using $\log(x + 1)$ after normalization and the RNA-sequencing pipeline run. Heatmaps were plotted using the ComplexHeatmap package (Gu et al, 2016). R statistical software version 4.0.3 was used for the statistical analyses and visualizations.

## cBioPortal database analyses

The correlation analysis between gene expression and breast cancer subtypes was examined using the METABRIC breast cancer cohort with PAM50 classification (Pereira et al, 2016), TCGA breast invasive carcinoma dataset and the TCGA Firehose legacy dataset (Gao et al, 2013). The RNA-seq data from breast invasive carcinoma samples and the relevant clinical information are available at the TCGA data portal, cBioPortal for Cancer Genomics (http://www.cbioportal.org/). According to the PAM50 classification, METABRIC breast cancer patients were divided into 5 subtypes, including the basal ($n = 199$), HER2 + ($n = 220$), Lum A ($n = 679$), Lum B ($n = 461$) and Normal-like ($n = 140$). ER and HER2 status were assessed using the patient's IHC information. The survival outcomes were extracted from the TCGA breast invasive carcinoma dataset.

For multivariable Cox regression analyses, $N = 121$ HER2+ patients from the TCGA-BRCA cohort were extracted, with reported pathological T-stages and lymph node statuses included as clinical factors and relapse-free survival modeled as the endpoint. Further, transcriptomics was included as log2-transformed values for RSEM normalized expression of DUSP6.

## Transcription factor binding analyses

To predict transcription factors enriched in the significantly differentially regulated genes during the therapy tolerance transitions we used ChEA3: transcription factor enrichment analysis by orthogonal omics integration tool (Keenan et al, 2019). Differentially regulated genes were provided as input to the tool and the significantly associated transcription factors were obtained using ENCODE (Encyclopedia of DNA Elements) ChIP-seq library. The output file consists of a list of transcription factors arranged based on a scaled rank where a lower value means a higher significance in the gene list. An FDR value < 0.05 was considered significant.

To predict the transcription factors binding the DUSP6 promotor region, Cistrome DB Toolkit was used (Zheng et al, 2019). As an input 4000 base pairs upstream of the DUSP6 gene were entered in the Cistrome DB Toolkit. This provided a list of

### The paper explained

**Problem**

The molecular mechanisms for the acquisition of resistance to HER2-targeted therapies in breast cancer are still elusive, especially when the drug-tolerant persister cells start to regrow under the treatment.

**Results**

The oncogenic phosphatase DUSP6 is increased in the drug-tolerant emerging persister cells and plays central roles in the acquisition of HER2i resistance. Moreover, DUSP6 blockade potentiates therapeutic sensitivity in the primary resistant models via inhibition of the HER3 signaling pathway. Inhibition of DUSP6 offers advantages over AKT blockade, including induction of apoptotic cell death, preempting ligand-induced rescue, and inhibition of HER3. Ultimately, DUSP6 inhibition reverses HER2i resistance in vivo and reduces the outgrowth of HER2+ cell in the brain metastasis models.

**Impact**

These findings have implications in the clinical management of HER2+ breast cancer and warrant development and clinical investigation of DUSP6 inhibitors in combination with HER2-directed therapies in the patients.

transcription factors based on the regulatory potential (RP score) derived by comparing the CHIP-seq data sets.

## Statistical analysis

Differentially expressed genes and pathways were identified using the R package limma (Ritchie et al, 2015). Latest hallmark gene sets were downloaded from the Molecular Signatures Database version 7.4 (http://www.gsea-msigdb.org/gsea/msigdb/index.jsp) and used within Gene Set Variation Analysis (GSVA) allowing pathway enrichment estimates for each sample (Hanzelmann et al, 2013), with FDR-cutoff <0.25 used for the pathway enrichment analyses. Data was transformed using $\log(x + 1)$ after normalization and the RNA-sequencing pipeline run. Heatmaps were plotted using the ComplexHeatmap package (Gu et al, 2016). R statistical software (R Core Team (2023). R: A language and environment for statistical computing. R Foundation for Statistical Computing, Vienna, Austria. URL: https://www.R-project.org) version 4.0.3 was used for the statistical analyses and visualizations.

All data were evaluated in triplicate against the vehicle-treated control cells and collected from three independent experiments. In addition to R analyses, data were visualized and analyzed using GraphPad Prism 8.3.0 using one-way ANOVA and the unpaired two-tailed student's $t$ test. All such data are presented as mean ± standard deviation (SD).

## Data availability

The gene expression data from this publication have been deposited to the GEO database (https://www.ncbi.nlm.nih.gov/geo/) and assigned the identifier GSE231526.

The source data of this paper are collected in the following database record: biostudies:S-SCDT-10_1038-S44321-024-00088-0.

## Peer review information

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

## Acknowledgements

Taina Kalevo-Mattila is acknowledged for superior technical support and the entire Turku Bioscience Centre personnel is thanked for excellent working environment. We acknowledge Zhong-Yin Zhang for PRL1i compound, Norma O'Donovan, National Institute for Cellular Biotechnology, Dublin City University, Ireland, for some HER2+ breast cancer cell lines used in this study, and Dihua Yu, The University of Texas MD Anderson Cancer Center, USA, for the BT474Br cells. We further acknowledge important contributions of the following core facilities of Turku Bioscience Centre (University of Turku and Åbo Akademi University) supported by Biocentre Finland: Finnish Functional Genomics Center, Screening unit, and Zebrafish unit. This study was supported by funding from Finnish Cancer Associations (JW), Foundation of Finnish Cancer Institute (MM), Maud Kuistila Foundation (MM), Turku University Foundation (MM), Finnish Cancer Institute (TDL), Finnish Cultural Foundation (TDL), and Finnish Cultural Foundation (KJK).

## Author contributions

**Majid Momeny**: Conceptualization; Data curation; Formal analysis; Funding acquisition; Investigation; Visualization; Writing—original draft.
**Mari Tienhaara**: Formal analysis; Investigation; Visualization. **Mukund Sharma**: Formal analysis; Investigation; Visualization. **Deepankar Chakroborty**: Investigation; Methodology. **Roosa Varjus**: Data curation; Formal analysis; Methodology. **Iina Takala**: Investigation. **Joni Merisaari**: Investigation; Methodology. **Artur Padzik**: Methodology. **Andreas Vogt**: Resources; Writing—review and editing. **Ilkka Paatero**: Formal analysis; Investigation; Methodology. **Klaus Elenius**: Resources; Supervision. **Teemu D Laajala**: Conceptualization; Data curation; Formal analysis; Investigation. **Kari J Kurppa**: Resources; Supervision; Visualization; Writing—review and editing. **Jukka Westermarck**: Conceptualization; Resources; Data curation; Supervision; Funding acquisition; Validation; Visualization; Writing—original draft; Project administration; Writing—review and editing.

Source data underlying figure panels in this paper may have individual authorship assigned. Where available, figure panel/source data authorship is listed in the following database record: biostudies:S-SCDT-10_1038-S44321-024-00088-0.

## Disclosure and competing interests statement

The authors declare no competing interests.

# Expanded View Figures

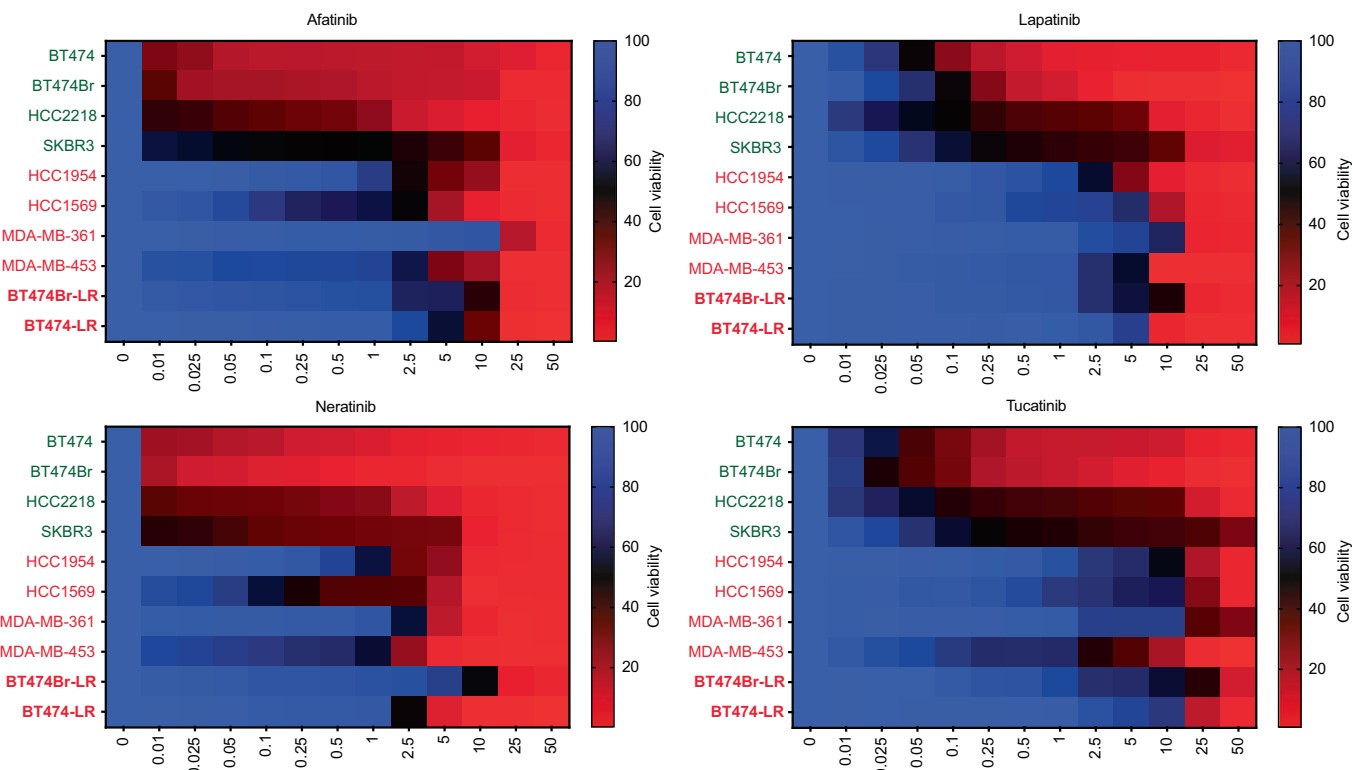

**Figure EV1.   Profiling of a panel of HER2+ breast cancer cell lines for their sensitivity to HER2 targeting small molecule tyrosine kinase inhibitors.**

The primary HER2i sensitive cells are marked on green, and the acquired HER2i resistant cells in red. The long-term resistant (LR) BT-474 and BT-474Br generated by 9-month treatment with lapatinib in this study are denoted in bold. The cells were treated with the increasing concentrations of the indicated HER2i compounds (in µM) for 48 h and cell viability was measured using WST-1 assay.

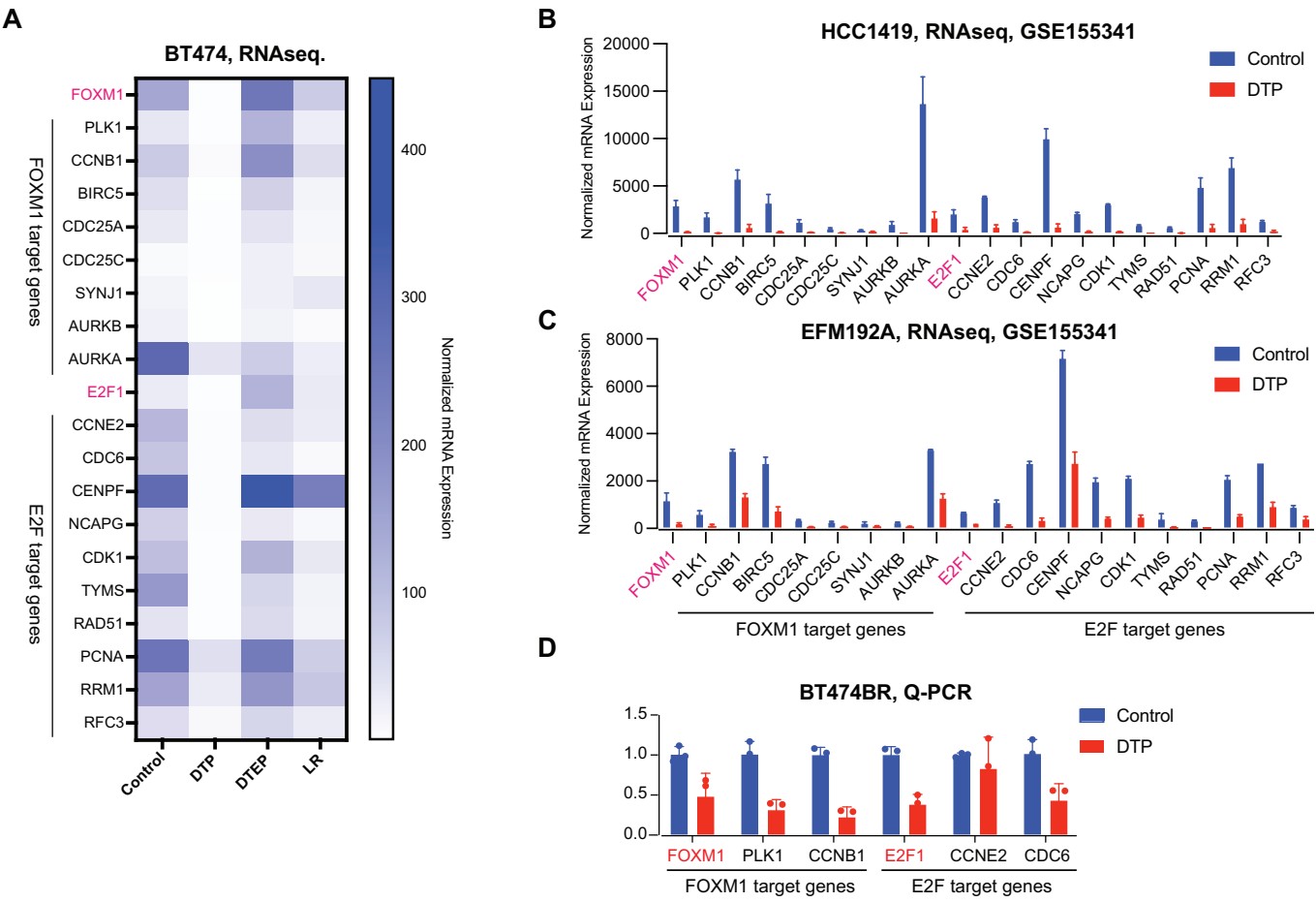

**Figure EV2. E2F1 and FOXM1 target genes are inhibited across HER2 + DTP cells.**

(A) mRNA levels of selected predicted FOXM1 and E2F1 target genes regulated during different steps of acquisition of lapatinib resistance in BT474 cells (Fig. 1). Data is blotted based on RNA sequencing analysis (Dataset EV1). (B, C) Expression of the predicted FOXM1 and E2F1 target genes in HER2 + HCC1419 and EFM192A cells between the DTP and the control cells. The RNA-seq data was obtained from Dataref: (Chang et al, 2022) (GSE155342). The cells were treated with lapatinib (2.5 μM) for 14 days to reach the DTP state. Data are shown as mean ± SD (n = 2). (D) Changes in the expression of FOXM1, PLK1, CCNB1, E2F1, CCNE2, and CDC6 in BT474Br cells between the control and the DTP cells treated for 9 days with lapatinib (1 μM). Data is based on Q-PCR analysis from three technical replicates. The analysis was limited to only these genes due to lack of sufficient mRNA material from the strongly growth suppressed DTP cells. Data are shown as mean ± SD (n = 3).

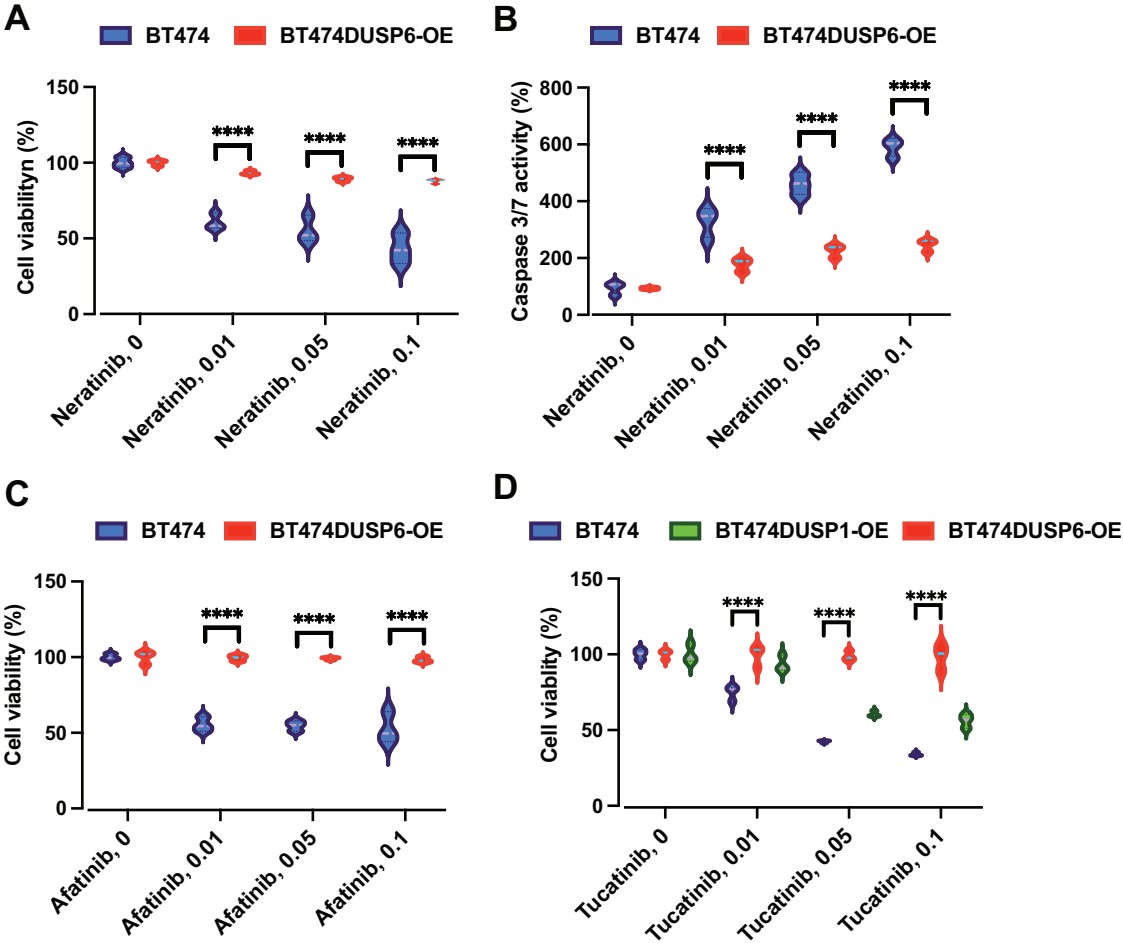

**Figure EV3.  DUSP6 overexpression protects HER2+ cells from HER2i-induced cell death.**

(**A, B**) Ectopic overexpression of DUSP6 in BT474 cells inhibits neratinib-elicited effects on cell viability and apoptosis, as measured by WST1 cell viability assay and caspase 3/7 activity, respectively. Data were collected from three independent experiments each performed in triplicate and analyzed by two-way ANOVA followed by Tukey' post hoc test. Statistically significant values of ****p < 0.0001 were determined. (**C**) Ectopic overexpression of DUSP6 in BT474 cells inhibits the afatinib-elicited effects on cell viability, as measured by WST1 assay. Data were collected from three independent experiments each performed in triplicate and analyzed by two-way ANOVA followed by Tukey' post hoc test. Statistically significant values of ****p < 0.0001 were determined. (**D**) Ectopic overexpression of DUSP6, but not of DUSP1, inhibits the Tucatinib-elicited effects on cell viability, as measured by WST1 assay. Data were collected from three independent experiments each performed in triplicate and analyzed by two-way ANOVA followed by Tukey' post hoc test. Statistically significant values of ****p < 0.0001 were determined.

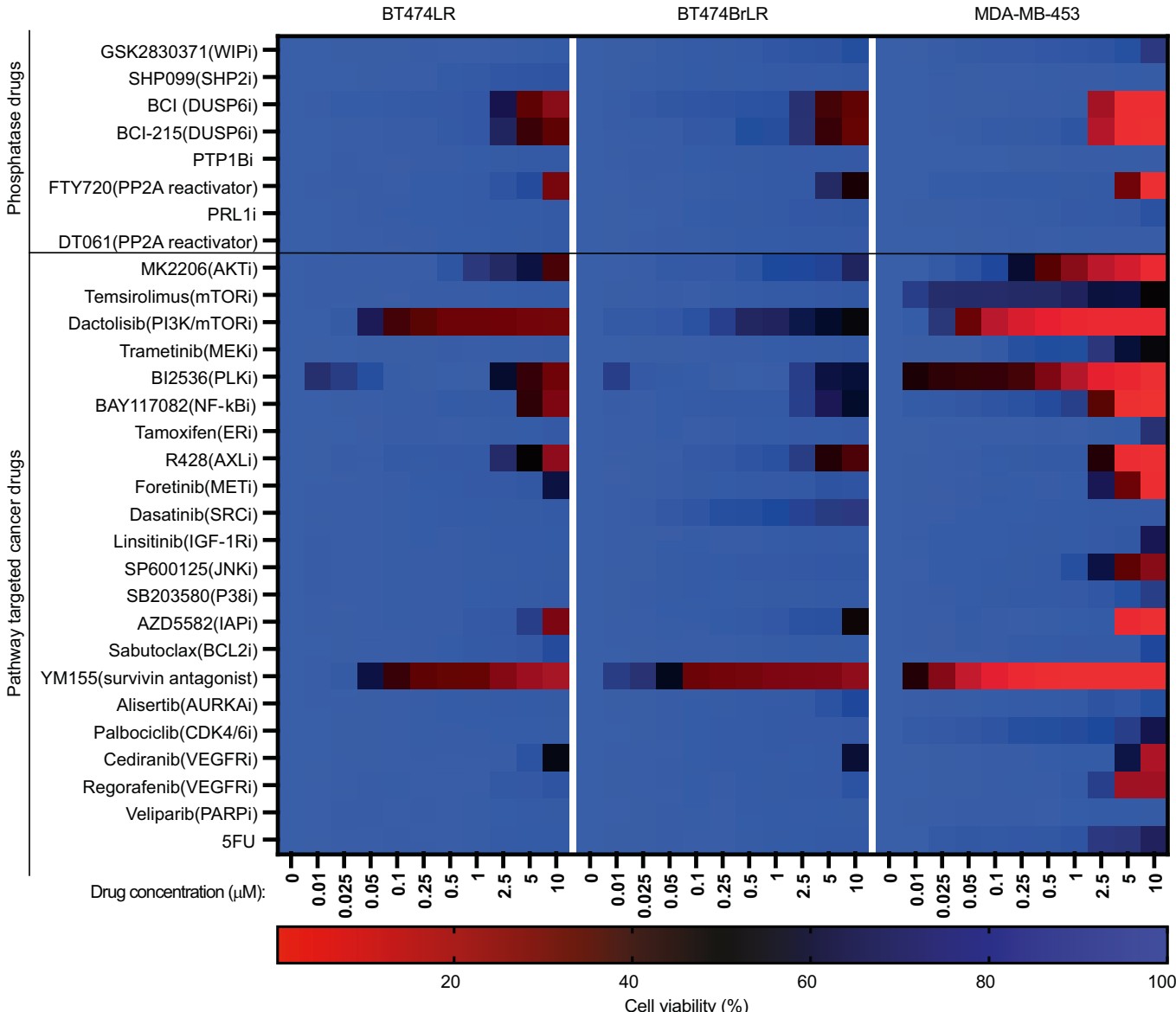

**Figure EV4. The anti-proliferative activities of a library of small molecule modulators of phosphatases, kinases, and anti-apoptotic proteins in HER2i-resistant cells.**

The indicated cells were treated with the increasing concentrations (in µM) of the compounds for 48 h and cell viability was measured using WST-1 assay. The long-term resistant (LR) BT474 and BT474Br generated de novo by 9-month treatment with lapatinib in this study had comparable drug sensitivity profile to acquired resistant cell line MDA-MB-453. The primary target of the used compound is indicated in parenthesis.

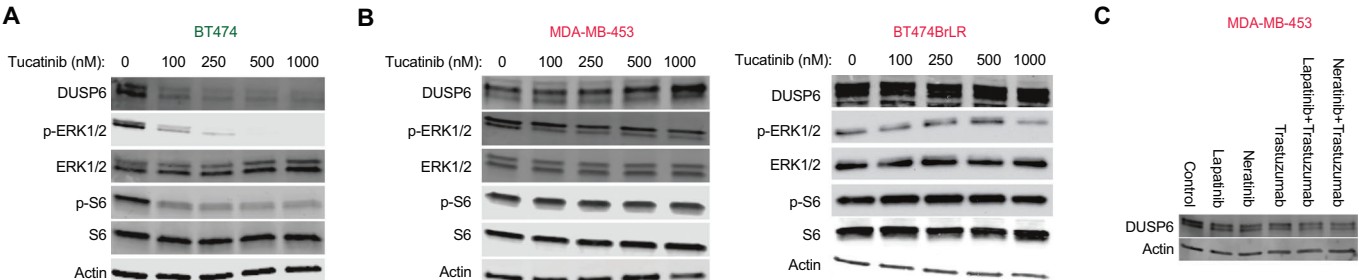

**Figure EV5.   Lack of DUSP6 inhibition is associated with HER2i resistance.**

(**A**, **B**) The effects of tucatinib on DUSP6 expression and the signaling pathway activities in (**A**) HER2i sensitive (green) BT474 cells or (**B**) HER2i resistant (red) MDA-MB-453 and BT474BrLR cells. The cells were treated with increasing concentrations of tucatinib for 48 h, followed by Western blot analysis. (**C**) DUSP6 expression in MDA-MB-453 cells is resistant to multiple HER2is including antibody therapy with trastuzumab. The cells were treated with indicated drugs for 48 h, followed by Western blot analysis.

