## [Peer Review File · EMBO Molecular Medicine]

DUSP6 inhibition overcomes Neuregulin/HER3-driven therapy tolerance in HER2+ breast cancer

Majid Momeny, Mari Tienhaara, Mukund Sharma, Deepankar Chakroborty, Roosa Varjus, Iina Takala, Joni Merisaari, Artur Padzik, Andreas Vogt, Ilkka Paatero, Klaus Elenius, Teemu Daniel Laajala, Kari Kurppa, and Jukka Westermarck

Corresponding authors: Jukka Westermarck (jukwes@utu.fi) , Majid Momeny (majid.momeny@uth.tmc.edu)

Review Timeline:

Submission Date:	4th Sep 23
Editorial Decision:	4th Sep 23
Revision Received:	8th Jan 24
Editorial Decision:	19th Jan 24
Revision Received:	8th May 24
Accepted:	24th May 24

Editor: Zeljko Durdevic

Transaction Report:

(Note: Please note that the manuscript was previously reviewed at another journal and the reports were taken into account in the decision making process at EMBO Molecular Medicine. Since the original reviews are not subject to EMBO's transparent review process policy, the reports and author response cannot be published.)

4th Sep 2023

Dear Prof. Westermarck,

Thank you for submitting your manuscript to EMBO Molecular Medicine. I have now carefully read your manuscript and point-by-point response to the referees' concerns and discussed them with the other members of our editorial team. We agreed that major revisions addressing the reviewers' concerns in full will be necessary for considering the manuscript in our journal.

EMBO Molecular Medicine encourages a single round of revision only and therefore, acceptance or rejection of the manuscript will depend on the completeness of your responses included in the next, final version of the manuscript. For this reason, and to save you from any frustrations in the end, I would strongly advise against returning an incomplete revision.

Further consideration of your manuscript will entail a second round of review and will depend on addressing the following points:

- Please pay particular attention to structure your manuscript (text and figures) in a more comprehensible manner.
- Provide clear rationale of the study.
- Include experiments with a rescue model and provide more mechanistic insight as suggested by referee #1.
- Provide data on toxicity and specificity of BCI.
- Single cell RNA sequencing and lineage tracing experiments are not required and should be discussed in the manuscript.
- For additional cell lines to validate the hypothesis, a 9 days DTP treatment of two other Lapatinib sensitive cells lines and qPCR validation of most prominent genes identified in BT474 cells is sufficient.

Additional experiments that further strengthen the main conclusions of the study are of course appreciated. We would welcome the submission of a revised version within three months for further consideration. Please let us know if you require longer to complete the revision.

I look forward to receiving your revised manuscript.

Yours sincerely,

Zeljko Durdevic

We require:

- 1) A .docx formatted version of the manuscript text (including legends for main figures, EV figures and tables). Please make sure that the changes are highlighted to be clearly visible.
- 2) Individual production quality figure files as .eps, .tif, .jpg (one file per figure). For guidance, download the 'Figure Guide PDF': (<https://www.embopress.org/page/journal/17574684/authorguide#figureformat>).
- 3) A .docx formatted letter INCLUDING the reviewers' reports and your detailed point-by-point responses to their comments. As part of the EMBO Press transparent editorial process, the point-by-point response is part of the Review Process File (RPF), which will be published alongside your paper.

4) A complete author checklist, which you can download from our author guidelines (<https://www.embopress.org/page/journal/17574684/authorguide#submissionofrevisions>). Please insert information in the checklist that is also reflected in the manuscript. The completed author checklist will also be part of the RPF.

6) It is mandatory to include a 'Data Availability' section after the Materials and Methods. Before submitting your revision, primary datasets produced in this study need to be deposited in an appropriate public database, and the accession numbers and database listed under 'Data Availability'. Please remember to provide a reviewer password if the datasets are not yet public (see <https://www.embopress.org/page/journal/17574684/authorguide#dataavailability>).

.

13) Author contributions: You will be asked to provide CRediT (Contributor Role Taxonomy) terms in the submission system. These replace a narrative author contribution section in the manuscript.

14) A Conflict of Interest statement should be provided in the main text.

Please note: When submitting your revision you will be prompted to enter your funding and payment information. This will allow Wiley to send you a quote for the article processing charge (APC) in case of acceptance. This quote takes into account any reduction or fee waivers that you may be eligible for. Authors do not need to pay any fees before their manuscript is accepted and transferred to the publisher.

EMBO Press participates in many Publish and Read agreements that allow authors to publish Open Access with reduced/no publication charges. Check your eligibility: <https://authorservices.wiley.com/author-resources/Journal-Authors/open-access/affiliation-policies-payments/index.html>

19th Jan 2024

Dear Prof. Westermarck,

Thank you for the submission of your revised manuscript to EMBO Molecular Medicine. We have now heard back from the one referee who we asked to evaluate your revised manuscript.

I have carefully read your manuscript, point-by-point response to the referees' comments, the referee report and discussed it with the other members of our editorial team. I am pleased to inform you that we will be able to accept your manuscript pending the following final amendments:

- 1) Please address all concerns raised by the referee. Point #1 and #3 should be addressed experimentally. I have asked the referee for additional clarification of the point #3 and we agreed that the xenograft experiment should be done with at least 2 cell lines in NOG mice.
- 2) Author checklist: Please submit a complete checklist. <https://www.embopress.org/pb-assets/embopress/EMBO%20Press%20Author%20Checklist-1642513524327.xlsx>
- 3) Please update the e-mail address of the co-corresponding author Majid Momeny to an institutional one.
- 4) Figures: We noted that some figures are in landscape format. Please check "Figure Guidelines" for more information about technical requirements and layout dimensions for figures. https://www.embopress.org/pb-assets/embopress/EMBOPress_Figure_Guidelines_061115-1561436025777.pdf
- 5) In the main manuscript file, please do the following:
 - Please address all comments suggested by our data editors listed below:
 - o Figure legends:
 1. Please indicate the statistical test used for data analysis in the legends of figures 2a-b.
 2. Please note that in figures 2c; 3d; 5b, d; there is a mismatch between the annotated p values in the figure legend and the annotated p values in the figure file that should be corrected.
 3. Please note that the box plot needs to be defined in terms of minima, maxima, centre, bounds of box and whiskers, and percentile in the legend of figure 2d.
 4. Please note that information related to n is missing in the legends of figures 2a-d, j; 3d; 5a-b, d; 6j; 7f-g; 8e; EV 2b-c.
 5. Please note that the error bars are not defined in the legends of figures 5a-b, d; EV 2b-d.
 - Add up to 5 keywords.
 - Rename "Methods" to "Materials and Methods".
 - Data availability: Please make sure that all data deposited in public repositories are freely accessible upon publication.
 - 6) Tables: Please rename Tables EV1 - 4 to Dataset EV1 - 4 and place their legends in a separate worksheet for the datasets. Please update the numbering for Tables EV5 - 7 to Tables EV1 -3 and all callouts in the main manuscript text.
 - 7) Appendix: Please add page numbers and correct nomenclature to "Appendix Figure S1" etc, also in the main manuscript text.
 - 8) Funding: Please make sure that information about all sources of funding are complete in both our submission system and in the manuscript. The Turku University Foundation and Finnish Cancer Institute are currently missing in our submission system.
 - 9) Synopsis:
 - Synopsis image: Please submit the visual abstract as a high-resolution jpeg file 550 px-wide x (250-400)-px high.
 - Please check your synopsis text and image before submission with your revised manuscript. Please be aware that in the proof stage minor corrections only are allowed (e.g., typos).
 - 10) For more information: This space should be used to list relevant web links for further consultation by our readers. Could you identify some relevant ones and provide such information as well? Some examples are patient associations, relevant databases, OMIM/proteins/genes links, author's websites, etc...
 - 11) As part of the EMBO Publications transparent editorial process initiative (see our Editorial at <http://embomolmed.embopress.org/content/2/9/329>), EMBO Molecular Medicine will publish online a Review Process File (RPF) to accompany accepted manuscripts. This file will be published in conjunction with your paper and will include the anonymous referee reports, your point-by-point response and all pertinent correspondence relating to the manuscript. Let us know whether you agree with the publication of the RPF and as here, if you want to remove or not any figures from it prior to publication. Please note that the Authors checklist will be published at the end of the RPF.
 - 12) Please provide a point-by-point letter INCLUDING my comments as well as the reviewer's reports and your detailed responses (as Word file).

I look forward to reading a new revised version of your manuscript as soon as possible.

Yours sincerely,

Zeljko Durdevic

*** Instructions to submit your revised manuscript ***

- 1) a .docx formatted version of the manuscript text (including Figure legends and tables)
- 2) Separate figure files*
- 3) supplemental information as Expanded View and/or Appendix. Please carefully check the authors guidelines for formatting Expanded view and Appendix figures and tables at <https://www.embopress.org/page/journal/17574684/authorguide#expandedview>
- 4) a letter INCLUDING the reviewer's reports and your detailed responses to their comments (as Word file).
- 5) The paper explained: EMBO Molecular Medicine articles are accompanied by a summary of the articles to emphasize the major findings in the paper and their medical implications for the non-specialist reader. Please provide a draft summary of your article highlighting
 - the medical issue you are addressing,
 - the results obtained and
 - their clinical impact.This may be edited to ensure that readers understand the significance and context of the research. Please refer to any of our published articles for an example.
- 6) For more information: There is space at the end of each article to list relevant web links for further consultation by our readers. Could you identify some relevant ones and provide such information as well? Some examples are patient associations, relevant databases, OMIM/proteins/genes links, author's websites, etc...
- 7) Author contributions: the contribution of every author must be detailed in a separate section.
- 8) EMBO Molecular Medicine now requires a complete author checklist (<https://www.embopress.org/page/journal/17574684/authorguide>) to be submitted with all revised manuscripts. Please use the checklist as guideline for the sort of information we need WITHIN the manuscript. The checklist should only be filled with page numbers where the information can be found. This is particularly important for animal reporting, antibody dilutions (missing) and exact values and n that should be indicated instead of a range.
- 9) Every published paper now includes a 'Synopsis' to further enhance discoverability. Synopses are displayed on the journal webpage and are freely accessible to all readers. They include a short stand first (maximum of 300 characters, including space) as well as 2-5 one sentence bullet points that summarise the paper. Please write the bullet points to summarise the key NEW findings. They should be designed to be complementary to the abstract - i.e. not repeat the same text. We encourage inclusion of key acronyms and quantitative information (maximum of 30 words / bullet point). Please use the passive voice. Please attach these in a separate file or send them by email, we will incorporate them accordingly.

You are also welcome to suggest a striking image or visual abstract to illustrate your article. If you do please provide a jpeg file

550 px-wide x 300-800px high.

10) A Conflict of Interest statement should be provided in the main text

11) Please note that we now mandate that all corresponding authors list an ORCID digital identifier. This takes <90 seconds to complete. We encourage all authors to supply an ORCID identifier, which will be linked to their name for unambiguous name identification.

Currently, our records indicate that the ORCID for your account is 0000-0001-7478-3018.

Link Not Available

Photos 400-800 DPI

*Additional important information regarding figures and illustrations can be found at

<https://bit.ly/EMBOPressFigurePreparationGuideline>. See also figure legend preparation guidelines:

<https://www.embopress.org/page/journal/17574684/authorguide#figureformat>

***** Reviewer's comments *****

Referee #1 (Comments on Novelty/Model System for Author):

see details below in comments to authors

Referee #1 (Remarks for Author):

This is a revised manuscript by Momeny and colleagues. The authors have responded to each of the specific comments of the reviewers and revised the manuscript accordingly. The revised manuscript is improved; however, some concerns remain.

1. The shDUSP6 with shRNA-resistant rescue is a very valid and standard request for shRNA experiments that was not addressed. Having multiple gRNAs, KO clones, and DUSP6 inhibitor is good, but not a definitive proof of specificity of the shRNA and not even the KO, especially not because the authors are using single cell clones. Thus, a rescue experiment in shRNA and KO cells is important.
2. For survival studies breast cancer patients have to be split to subtypes, since subtypes impact survival. It appears that for the KM plots in Fig 2G-H the authors focused on HER2+ breast cancer, but this is not obvious in the figure. In Fig 2E-F also the data is presented in an unusual way. To rigorously address if DUSP6 expression has a significant impact on the clinical outcome of HER2+ breast cancer patients and if this is specific to HER2+ breast cancer, the authors should perform multivariate regression analyses.
3. Different experiments are done with different cell line models making it difficult to combine all data into one mechanistic model. Key experiments like xenograft studies have to be performed by multiple cell lines. especially because the authors used very different HER2+ models: BT474 is ER+ luminal ERBB2 amplified, HCC1954 is basal ERBB2 amplified, MDA-MB-453 is luminal ERBB2 non-amplified (just gained).

Response to reviewers

We are very grateful for the constructive criticism from the reviewer towards our work. The manuscript has now been revised according to reviewer's comments and we have added all requested new evidence to support the main conclusions of the study. We have also rewritten the text and restructured the figures to more clearly present the main novelties and discoveries included in the work. We sincerely hope that the manuscript can be accepted in its present form to be published in the *Embo Molecular Medicine*.

Reviewer comments:

1. The shDUSP6 with shRNA-resistant rescue is a very valid and standard request for shRNA experiments that was not addressed. Having multiple gRNAs, KO clones, and DUSP6 inhibitor is good, but not a definitive proof of specificity of the shRNA and not even the KO, especially not because the authors are using single cell clones. Thus, a rescue experiment in shRNA and KO cells is important.

Response: In order to respond to this request, we cloned a lentiviral DUSP6 expression vector and used it to overexpress DUSP6 in one of the DUSP6 single cell KO clones (Cl. 6, g2). The expression levels in the antibiotic resistant rescue clones reached about 50% of endogenous DUSP6 levels as compared to CAS9 expressing control cells (panel B). Regardless of suboptimal expression, the lentivirally expressed DUSP6 significantly rescued the KO clone colony growth (panels C and D). When colony growth was normalized to DUSP6 protein levels between the CAS9 expressing control cells and the rescue clones, the colony growth potential was indistinguishable between the cell lines. Therefore this data provides the requested evidence that the lack of colony growth in Cripsr/CAS9 targeted DUSP6 KO cells was due to loss of DUSP6. This data is shown now in **New Appendix Figure S4**.

2. For survival studies breast cancer patients have to be split to subtypes, since subtypes impact survival.

Response Using the same TCGA dataset from which the ERBB2high (HER2+) subtype was analysed (Current Figs. 2F,G), we now present survival analysis from Luminal B, and basal type subtypes, as well as from all breast cancer subtypes combined (**New Appendix Figure S2E**). As a conclusion, high DUSP6 expression was significant predictor of poor survival only in HER2+ subtype. The data is described in page 13.

It appears that for the KM plots in Fig 2G-H the authors focused on HER2+ breast cancer, but this is not obvious in the figure.

Response: The figure has been adjusted (Current Figs. 2F,G)

In Fig 2E-F also the data is presented in an unusual way.

Response: This was one of the three options in which the data could be exported from the TCGA and we evaluated it to be the most easily understandable format. No raw data is available for own figure assembly.

To rigorously address if DUSP6 expression has a significant impact on the clinical outcome of HER2+ breast cancer patients and if this is specific to HER2+ breast cancer, the authors should perform multivariate regression analyses.

Response: We have now performed multivariate regression analysis and the results show that increase in DUSP6 expression and large tumor size (T4) remained significant independent prognostic factors (**New Appendix Figure S2D**). The data is described in page 13.

3. Different experiments are done with different cell line models making it difficult to combine all data into one mechanistic model. Key experiments like xenograft studies have to be performed by multiple cell lines. especially because the authors used very different HER2+ models: BT474 is ER+ luminal ERBB2 amplified, HCC1954 is basal ERBB2 amplified, MDA-MB-453 is luminal ERBB2 non-amplified (just gained).

Response: This comment is very valid but was already addressed in the original version of the manuscript. The xenograft data in figure 5 was done by using two different cell lines MDA-MB-453 (Fig. 5A,D,E) and HCC1954 (Fig. 5 B,C). In addition, we used two different HER2 inhibitors, Lapatinib and Neratinib, as a well as both genetic and pharmacological inhibition of DUSP6. In addition, in Fig. 7G, we used yet another HER2i resistant cell line MDA-MB-361 in xenograft experiment to demonstrate therapeutical potential of DUSP6 targeting. Since results in all these experiments, using three cell lines, support the same conclusion that DUSP6 inhibition targets HER2i resistance, we found this data very convincing, and fully addressing this concern. We agree that the cell line independence of the results in figure 5 might have been better emphasised in the text, and have re-written the manuscript and figure legends in those regards.

Editor comments:

3) Please update the e-mail address of the co-corresponding author Majid Momeny to an institutional one.

Response: Revised as instructed.

4) Figures: We noted that some figures are in landscape format. Please check "Figure Guidelines" for more information about technical requirements and layout dimensions for figures. https://www.embopress.org/pb-assets/embo-site/EMBOPress_Figure_Guidelines_061115-1561436025777.pdf

Response: Revised as instructed.

5) In the main manuscript file, please do the following:

- Please address all comments suggested by our data editors listed below:

o Figure legends:

1. Please indicate the statistical test used for data analysis in the legends of figures 2a-b.

Response: Revised as instructed.

2. Please note that in figures 2c; 3d; 5b, d; there is a mismatch between the annotated p values in the figure legend and the annotated p values in the figure file that should be corrected.

Response: Revised as instructed.

3. Please note that the box plot needs to be defined in terms of minima, maxima, centre, bounds of box and whiskers, and percentile in the legend of figure 2d.

Response: Revised as instructed.

4. Please note that information related to n is missing in the legends of figures 2a-d, j; 3d; 5a-b, d; 6j; 7f-g; 8e; EV 2b-c.

Response: Revised as instructed.

5. Please note that the error bars are not defined in the legends of figures 5a-b, d; EV 2b-d.

- Add up to 5 keywords.

- Rename "Methods" to "Materials and Methods".

- Data availability: Please make sure that all data deposited in public repositories are freely accessible upon publication.

Response: Revised as instructed.

6) Tables: Please rename Tables EV1 - 4 to Dataset EV1 - 4 and place their legends in a separate worksheet for the datasets. Please update the numbering for Tables EV5 - 7 to Tables EV1 -3 and all callouts in the main manuscript text.

Response: Revised as instructed.

7) Appendix: Please add page numbers and correct nomenclature to "Appendix Figure S1" etc, also in the main manuscript text.

Response: Revised as instructed.

8) Funding: Please make sure that information about all sources of funding are complete in both our submission system and in the manuscript. The Turku University Foundation and Finnish Cancer Institute are currently missing in our submission system.

9) Synopsis:

- Synopsis image: Please submit the visual abstract as a high-resolution jpeg file 550 px-wide x (250-400)-px high.

Response: Revised as instructed.

11) As part of the EMBO Publications transparent editorial process initiative (see our Editorial at <http://embomolmed.embopress.org/content/2/9/329>), EMBO Molecular Medicine will publish online a Review Process File (RPF) to accompany accepted manuscripts. This file will be published in conjunction with your paper and will include the anonymous referee reports, your point-by-point response and all pertinent correspondence relating to the manuscript. Let us know whether you agree with the publication of the RPF and as here, if you want to remove or not any figures from it prior to publication. Please note that the Authors checklist will be published at the end of the RPF.

Response: This is OK for us.

24th May 2024

Dear Prof. Westermarck,

We are pleased to inform you that your manuscript is accepted for publication and is now being sent to our publisher to be included in the next available issue of EMBO Molecular Medicine.
